# Conditional Coverage Estimation for High-quality Prediction Intervals

## Abstract

Deep learning has achieved state-of-the-art performance to generate high-quality prediction intervals (PIs) for uncertainty quantification in regression tasks. The high-quality criterion requires PIs to be as narrow as possible, whilst maintaining a pre-specified level of data (marginal) coverage. However, most existing works for high-quality PIs lack accurate information on conditional coverage, which may cause unreliable predictions if it is significantly smaller than the marginal coverage. To address this problem, we propose a novel end-to-end framework which could output high-quality PIs and *simultaneously* provide their conditional coverage estimation. In doing so, we design a new loss function that is both easy-to-implement and theoretically justified via an exponential concentration bound. Our evaluation on real-world benchmark datasets and synthetic examples shows that our approach not only outperforms the state-of-the-arts on high-quality PIs in terms of average PI width, but also accurately estimates conditional coverage information that is useful in assessing model uncertainty.

## 1 Introduction

Prediction interval (PI) is poised to play an increasingly prominent role in uncertainty quantification for regression tasks (Khosravi et al., 2010; 2011; Galván et al., 2017; Rosenfeld et al., 2018; Tagasovska & Lopez-Paz, 2018; 2019; Romano et al., 2019; Wang et al., 2019; Kivaranovic et al., 2020). A high-quality PI should be as narrow as possible, whilst maintaining a pre-specified level of data coverage or marginal coverage (Pearce et al., 2018). Compared with PIs obtained based on coverage-only consideration, the "high-quality" criterion is beneficial in balancing between marginal coverage probability and interval width. However, the conditional coverage given a feature, which is critical for making reliable context-based decisions, is unassessed and missing in most existing works on high-quality PIs. In the presence of heteroskedasticity and model misspecification, the marginal coverage can be very different from the conditional coverage at a given point, which affects the downstream decision-making task that relies on the uncertainty information provided by the PI. Our main goal is to meaningfully incorporate and assess conditional coverages in high-quality PIs.

Conditional coverage estimation is challenging for two reasons. First is that the natural evaluation metric of *conditional coverage error*, an $L^p$ distance between the estimated and ground-truth conditional coverages, is difficult to compute as it requires obtaining the conditional probability given feature $x$, which is arguably as challenging as the regression problem itself. Our first goal in this paper is to address this issue by developing a new metric called *calibration-based conditional coverage error* for conditional coverage estimation measurement. Our approach is inspired from the calibration notion in classification (Guo et al., 2017). The basic idea is to relax conditional coverage at any given point to being averaged over all points that bear the same estimated value. An estimator satisfying the relaxed property is regarded as well-calibrated. In regression, calibration-based conditional coverage error provides a middle ground between the enforcement of marginal coverage (lacking any conditional information) and conditional coverage (computationally intractable). Compared with conditional coverage, this middle-ground metric can be viewed as a "dimension reduction" of the conditioning variable from the original sample space to the space $[0, 1]$, so that we can easily discretize to compute the empirical metric values.

The second challenge is the discontinuity in the above metrics that hinders efficient training of PIs that are both high-quality and possess reliable conditional coverage information. To address this, we design a new loss function based on a combination of the high-quality criterion and a *coverage assessment loss*. The latter can be flexibly added as a separate module to any neural network (NN) used to train PIs. It is based on an empirical version of a tight upper bound on the coverage error in terms of a Kullback–Leibler (KL) divergence, which can be readily employed for running gradient descent. We theoretically show how training with our proposed loss function attains this upper-bounding value via a concentration bound. We also demonstrate the empirical performance of our approach in terms of PI quality and conditional coverage assessment compared with benchmark methods.

**Summary of Contributions**: (1) We identify the conditional coverage estimation problem as a new challenge for high-quality PIs and introduce a new evaluation metric for coverage estimation. (2) We propose an *end-to-end* algorithm that can simultaneously construct high-quality PIs and generate conditional coverage estimates. In addition, we provide theoretical justifications on the effectiveness of our algorithm by developing concentration bounds relating the coverage assessment loss and conditional coverage error. (3) By evaluating on benchmark datasets and synthetic examples, we empirically demonstrate that our approach not only achieves high performance on conditional coverage estimation, but also outperforms the state-of-the-art algorithms on high-quality PI generation.

## 2 EVALUATING CONDITIONAL COVERAGE FOR HIGH-QUALITY PIS

Let $X \in \mathcal{X}$ and $Y \in \mathcal{Y} \subset \mathbb{R}$ be random variables denoting the input feature and label, where the pair $(X, Y)$ follows an (unknown) ground-truth joint distribution $\pi(X, Y)$. Let $\pi(Y|X)$ be the conditional distribution of $Y$ given $X$. We are given the training data $\mathcal{D} := \{(x_i, y_i), \ i = 1, 2, \cdots, n\}$ where $(x_i, y_i)$ are i.i.d. realizations of random variables $(X, Y)$. A PI refers to an interval $[L(x), U(x)]$ where $L$, $U$ are two functions mapping from $\mathcal{X}$ to $\mathcal{Y}$ trained on the data $\mathcal{D}$. $[L(x), U(x)]$ is called a PI at *prediction level* $1 - \alpha$ ($0 \leq \alpha \leq 1$) if its *marginal coverage* is not less than $1 - \alpha$, i.e., $\mathbb{P}[Y \in [L(X), U(X)]|L, U] \geq 1 - \alpha$ where $\mathbb{P}$ is with respect to a new test point $(X, Y) \sim \pi$.

We say that $[L(x), U(x)]$ is of high-quality if its marginal coverage attains a pre-specified target prediction level and has a short width on average. In particular, a best-quality PI at prediction level $1 - \alpha$ is an optimal solution to the following constrained optimization problem:

$$\min_{L,U} \ \mathbb{E}[U(X) - L(X)] \quad \text{subject to} \quad \mathbb{P}[Y \in [L(X), U(X)]|L, U] \geq 1 - \alpha. \qquad (2.1)$$

The high-quality criterion has been widely adopted in previous work (see Section 6). However, this criterion alone may fail to carry important model uncertainty information at specific test points. Consider a simple example where $x \sim \text{Uniform}[0, 1]$, $y = 0$ for $x \in [0, 0.95]$ and $y|x \sim \text{Uniform}[0, 1]$ for $x \in (0.95, 1]$. Then according to equation 2.1, a best-quality 95% PI is precisely $L(x) = U(x) = 0$ for all $x \in [0, 1]$. This PI has nonconstant coverage if we condition at different points (1 for $x \in [0, 0.95]$ and 0 for $x \in (0.95, 1]$), and can deviate significantly from the overall coverage 95%. More examples to highlight the need of obtaining conditional coverage information can be found in our numerical experiments in Section 5.1.

To mitigate the drawback of the high-quality criterion, we define:

**Definition 2.1** (Conditional Coverage and Its Estimator). *The conditional coverage associated with a PI $[L(x), U(x)]$ is $A(x) := \mathbb{P}[Y \in [L(X), U(X)]|L, U, X = x]$ for a.e. $x \in \mathcal{X}$, where $\mathbb{P}$ is taken with respect to $\pi(Y|X)$. For a (conditional) coverage estimator $\hat{P}$, which is a measurable function from $\mathcal{X}$ to $[0, 1]$, we define its $L^p$ conditional coverage error ($\widetilde{CE}_p$) as*

$$\widetilde{CE}_p := \left\| \mathbb{P}[Y \in [L(X), U(X)]|L, U, X] - \hat{P}(X) \right\|_{L^p(\mathcal{X})}$$

*where the $L^p$-norm is taken with respect to the randomness of $X$ ($1 \leq p \leq +\infty$).*

Note that evaluating $\widetilde{CE}_p$ relies on approximating the conditional coverage $A(x)$, which can be as challenging as the original prediction problem. To address this, we leverage the similarity of estimating $A(x)$ to generating prediction probabilities in binary classification, which motivates us to

borrow the notion of calibration in classification. This idea is based on a relaxed error criterion by looking at the conditional coverage among all points that bear the same coverage estimator value, instead of conditioning at any given point. The resulting error metric then only relies on probabilities conditioned on variables in a much lower-dimensional space $[0, 1]$ than $\mathcal{X}$. To explain concretely, we introduce a "perfect-calibrated coverage estimator" as:

**Definition 2.2** (Perfect Calibration). *A coverage estimator $\hat{P}$ is called a perfect-calibrated coverage estimator associated with $[L(x), U(x)]$ if it satisfies*

$$\hat{P}(x) = \mathbb{P}[Y \in [L(X), U(X)]|L, U, \hat{P}(X) = \hat{P}(x)], \quad a.e. \ \hat{P}(x) \in [0, 1]. \tag{2.2}$$

*where $a.e.$ is with respect to the probability measure on $[0, 1]$ induced by the random variable $\hat{P}(X)$.*

Equation 2.2 means that a point $x$ with conditional coverage estimate $\hat{P}(x) = p$ has an average coverage of precisely $p$, among all points in $\mathcal{X}$ that possess the same conditional coverage estimated value. That is, the average coverage of the PI restricted on the subset $\{x \in \mathcal{X} : \hat{P}(x) = p\}$ should be precisely $p$. Corresponding to Definition 2.2, we define:

**Definition 2.3** (Calibration-based Error). *An $L^p$ ($1 \leq p \leq +\infty$) calibration-based conditional coverage error, or coverage error for short ($CE_p$), of a coverage estimator $\hat{P}$ is:*

$$CE_p := \left\| \mathbb{P}[Y \in [L(X), U(X)]|L, U, \hat{P}(X)] - \hat{P}(X) \right\|_{L^p(\mathcal{X})} \tag{2.3}$$

*where $L^p$-norm is taken with respect to the randomness of $\hat{P}(X)$.*

In the above definition the conditional probability $\mathbb{P}[Y \in [L(X), U(X)]|L, U, \hat{P}(X)]$ is a measurable function of random variable $\hat{P}(X)$, say $\gamma(\hat{P}(X))$. By a change of variable,

$$CE_p^p := \left\| \gamma(\hat{P}(X)) - \hat{P}(X) \right\|_{L^p(\mathcal{X})}^p = \int_0^1 |\gamma(t) - t|^p dF_{\hat{P}(X)}(t) \tag{2.4}$$

where $F_{\hat{P}(X)}$ is a probability distribution of $\hat{P}(X)$ on $[0, 1]$. Here, $CE_p$ only requires estimating $\gamma(t)$ for $t \in [0, 1]$, which can be done easily by discretizing $[0, 1]$ for empirical calculation. We call the empirical $L^p$ calibration-based conditional coverage error $ECE_p$. More details of this empirical error can be found in Appendix A.4.

A calibration-based error $CE_p$ provides a middle ground between the enforcement of marginal coverage and conditional coverage. The ground-truth conditional coverage is perfectly calibrated, but not vice versa. However, if we enforce the perfect calibration criterion for a coverage estimator to hold when restricted to any measurable subset in $\mathcal{X}$, then the choice of estimator will reduce uniquely to the conditional coverage (Definition A.1 and Lemma A.2). In this sense, $CE_p$ is an error metric that is a natural relaxation of $\widetilde{CE_p}$, and although less precise, $CE_p$ is computationally much more tractable than $\widetilde{CE_p}$.

**Evaluation Metric for Coverage Estimator.** We use $ECE_1$ as the primary evaluation metric to measure the quality of a coverage estimator. A high $ECE_1$ value of a coverage estimator indicates an unreliable coverage estimation while a small $ECE_1$ value indicates that the coverage estimator is close to the perfect-calibrated property. Ideally, an effective algorithm should output a coverage estimator with a small $ECE_1$ value.

## 3 METHODOLOGY

We propose a novel end-to-end algorithm, named coverage assessment network (CaNet), to simultaneously generate a coverage estimator along with the high-quality PI. As illustrated in Figure 1, our CaNet consists of two major modules: (1) **predictor module** and (2) **coverage assessment module** (Ca-Module). The predictor module provides the upper and lower bound of the estimated PIs. Meanwhile, the Ca-Module is added to the output layer to access the conditional coverage information of PIs from the predictor module. Our model is jointly optimized by three loss functions: coverage assessment loss $L_{CA}$, intervals width loss $L_{IW}$, and coverage probability loss $L_{CP}$. Benefited from these powerful modules, the CaNet can *generate* and *validate* the coverage estimator at the same time without any requirement for further post-processing steps.

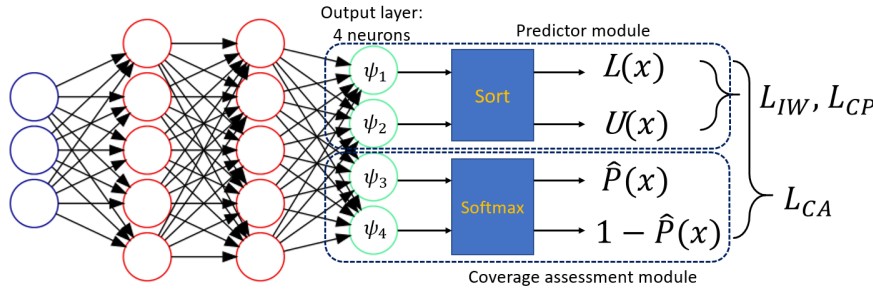

Figure 1: The framework of our proposed coverage assessment network (CaNet).

**Coverage Assessment Module**. Our Ca-Module consists of two neurons fully connected to the last hidden layer to estimate the conditional coverage. After passing through the softmax activation function, it outputs a two-point probability distribution $\hat{P}(x), 1 - \hat{P}(x)$ where $\hat{P}(x)$ is the coverage estimator of the PI from the predictor module. Our Ca-Module can be easily integrated into the output layer of deep neural networks to estimate their conditional coverage.

**Loss Function Design and Tuning Procedure.** Our loss function is a sum of the *predictor loss* and the *coverage loss*. The predictor loss aims to narrow the prediction intervals as much as possible, while maintaining a specified marginal coverage of data. Inspired by (Khosravi et al., 2010; 2011; Pearce et al., 2018; Rosenfeld et al., 2018), our predictor loss is formed by the sum of intervals width (IW) loss $L_{IW}$ and coverage probability (CP) loss $L_{CP}$:

$$L_{IW} = \frac{1}{n} \sum_{i=1}^{n} (U(x_i) - L(x_i)), \tag{3.1}$$

$$L_{CP} = \frac{1}{n} \sum_{i=1}^{n} \tilde{k}_i, \quad CP = \frac{1}{n} \sum_{i=1}^{n} k_i, \tag{3.2}$$

where $k_i$ indicates whether each data point has been captured by the PIs: $k_i = 1$ if $L(x_i) \leq y_i \leq U(x_i)$ and $k_i = 0$ otherwise. $\tilde{k}_i$ is a soft version of $k_i$, which is defined as: $\tilde{k}_i := \sigma(\lambda_3(U(x_i) - y_i)) \cdot \sigma(\lambda_3(y_i - L(x_i)))$, where $\lambda_3 \geq 0$ is a tunable parameter and $\sigma(t) := \frac{1}{1+e^{-t}}$ is the sigmoid function. Therefore, $L_{CP}$ is a soft version of $CP$ that can be used for gradient descent. Associated with the Ca-Module, we introduce a coverage assessment loss $L_{CA}$ to estimate the conditional coverage:

$$L_{CA} = \frac{1}{n} \sum_{i=1}^{n} \left( k_i \log(\hat{P}(x_i)) + (1 - k_i) \log(1 - \hat{P}(x_i)) \right) \tag{3.3}$$

We will show in Section 4 that the expectation of coverage assessment loss $L_{CA}$ provides an upper bound for both the conditional coverage error (Definition 2.1) and calibration-based conditional coverage error (Definition 2.3). Hence, minimizing $L_{CA}$ contributes to the recovery of the conditional coverage. In order to run gradient-based methods, we replace the discrete indicator $(k_i, 1 - k_i)$ in $L_{CA}$ with its soft version $(\tilde{k}_i, 1 - \tilde{k}_i)$:

$$\tilde{L}_{CA} = \frac{1}{n} \sum_{i=1}^{n} \left( \tilde{k}_i \log(\hat{P}(x_i)) + (1 - \tilde{k}_i) \log(1 - \hat{P}(x_i)) \right) \tag{3.4}$$

Our total loss function for the CaNet is defined as:

$$\text{Total Loss} = L_{IW} + \lambda_1(1 - L_{CP}) + \lambda_2 \tilde{L}_{CA} \tag{3.5}$$

where $\lambda_1 \geq 0, \lambda_2 \geq 0$ are tunable parameters. We propose an easy-to-implement yet effective tuning procedure to pick up these parameters. Please refer to Appendix D for more algorithm details.

**Deep Ensembles.** Following previous research (Lee et al., 2015; Lakshminarayanan et al., 2017; Pearce et al., 2018; Fort et al., 2019; Ovadia et al., 2019; Gustafsson et al., 2020; Pearce et al., 2020), we apply the deep ensemble technique to provide more robust and better results. During the training period, with the same hyperparameters $\lambda_i, i = 1, 2, 3$, $m$ networks are trained with different

initializations. The prediction results from $i$-th network are denoted as: $([L_i(x), U_i(x)], \hat{P}_i(x), 1 - \hat{P}_i(x))$. Finally, the output from CaNet is:

$$
\begin{array}{llll}
\textit{Lower bound} & \bar{L} & := & \sum_{i=1}^m \frac{1}{m} L_i, \\
\textit{Upper bound} & \bar{U} & := & \sum_{i=1}^m \frac{1}{m} U_i, \\
\textit{Coverage estimator} & \hat{\bar{P}} & := & \sum_{i=1}^m \frac{1}{m} \hat{P}_i.
\end{array}
\tag{3.6}
$$

## 4 THEORETICAL ANALYSIS

In this section, we provide theoretical insights that minimizing $L_{CA}$ is equivalent to minimizing a tight upper bound of the conditional coverage error and thus can recover the true conditional coverage. To achieve this, we first prove that both $CE_p$ and $\widetilde{CE}_p$ are bounded above by the expectation of a Kullback–Leibler divergence-type random variable $K_1(X)$. Then we show that $L_{CA}$ is an empirical counterpart of $K_1(X)$ and establish a concentration bound between $L_{CA}$ and $\mathbb{E}[K_1(X)]$.

**Theorem 4.1.** *Let $A(x) := \mathbb{P}[Y \in [L(X), U(X)] | L, U, X = x]$ be the conditional coverage in Definition 2.1. Let $K(x) = A(x) \log\left(\frac{A(x)}{\hat{P}(x)}\right) + (1 - A(x)) \log\left(\frac{1 - A(x)}{1 - \hat{P}(x)}\right)$. Then*

$$
CE_p \leq \widetilde{CE}_p \leq \left(\frac{1}{2} \mathbb{E}[K(X)]\right)^{\alpha_p/2}, \forall 1 \leq p \leq +\infty
$$

*where $\alpha_p = 1, \forall 1 \leq p \leq 2$ and $\alpha_p = \frac{2}{p}, \forall 2 \leq p \leq +\infty$. Moreover, the inequality is attainable if, e.g., $\hat{P}(x)$ equals the conditional coverage $A(x)$.*

From Theorem 4.1, we see that minimizing $\mathbb{E}[K(x)]$ is equivalent to minimizing a tight upper bound for the coverage error. For every $x$, $K(x)$ is the Kullback–Leibler divergence between the distributions represented by $(\hat{P}(x), 1 - \hat{P}(x))$ and $(A(x), 1 - A(x))$. $K(x) = K_0(x) + K_1(x)$, where $K_0(x) = A(x) \log(A(x)) + (1 - A(x)) \log(1 - A(x))$ and $K_1(x) = -A(x) \log(\hat{P}(x)) - (1 - A(x)) \log(1 - \hat{P}(x))$. Minimizing $\mathbb{E}[K(X)]$ over $\hat{P}$ is equivalent to minimizing $\mathbb{E}[K_1(X)]$. The type of results in Theorem 4.1 that bounds an $L^p$ conditional coverage error via a Kullback-Leibler-type error is new as far as we know. Next, to show $L_{CA}$ approximates $\mathbb{E}[K_1(X)]$, we need the following assumptions:

**Assumption 4.2.** *The four classes of functions $([L(x), U(x)], \hat{P}(x), 1 - \hat{P}(x))$ output by the neural network (NN) in Figure 1 have finite VC dimensions, say they are bounded above by $V_0$.*

Assumption 4.2 holds for a wide range of NNs (e.g., Theorem 8.14 in Anthony & Bartlett (2009), Theorem 7 in Bartlett et al. (2019)). In particular, it holds for the one we adopt in the experiments (where we use the ReLU-activated NN to construct $\psi_i$, $i = 1, 2, 3, 4$; see Section 5):

**Theorem 4.3.** *Suppose $\psi_i$, $i = 1, 2, 3, 4$ are the pre-activated output neurons of the NN in Figure 1 using the ReLU activation function. Then Assumption 4.2 holds. Moreover, suppose the NN has $W$ parameters and $U$ computation units (nodes). Then $V_0 = O(WU)$.*

**Assumption 4.4.** *$|\log(\hat{P}(x))| \leq M$, $|\log(1 - \hat{P}(x))| \leq M$ for all $x$ and $\hat{P}$.*

This is a natural assumption in practice because $\log(\hat{P}(x))$ and $\log(1 - \hat{P}(x))$ are replaced by $\log(\hat{P}(x) + \epsilon)$ and $\log(1 - \hat{P}(x) + \epsilon)$ respectively to avoid explosion when implementing the algorithm. In particular, in our experiments in Section 5, $\epsilon = 0.1^6$ and thus $M = 14$. Let

$$
\mathcal{F} = \{f(x, y) = I_{y \in [L(x), U(x)]} : L, U \text{ are output by the NN}\},
$$

$$
\mathcal{G} = \{\hat{P}(x) : \hat{P} \text{ is output by the NN}\}.
$$

**Theorem 4.5.** *Suppose Assumptions 4.2 and 4.4 hold. The training data $\mathcal{D} = \{(x_i, y_i), \ i = 1, 2, \cdots, n\}$ where $(x_i, y_i)$ are i.i.d. samples $\sim \pi$. Recall that the (hard) coverage estimator assessment loss is $L_{CA} = -\frac{1}{n} \sum_{i=1}^n \left(f(x_i, y_i) \log(\hat{P}(x_i)) + (1 - f(x_i, y_i)) \log(1 - \hat{P}(x_i))\right)$. Then for any $t > 0$, we have*

$$
\mathbb{P}\left(\sup_{f \in \mathcal{F}, \hat{P} \in \mathcal{G}} |L_{CA} - \mathbb{E}[K_1(X)]| \geq t\right) \leq C^* e^{-\frac{nt^2}{16M^2}}.
$$

*where $C^*$ only depends on $V_0$ in Assumption 4.2.*

Theorem 4.5 shows that the coverage assessment loss approximates $\mathbb{E}[K_1(x)]$ well with an exponential tail bound. The difficulty in analyzing Theorem 4.5 lies in the fact that the hypothesis classes in Assumption 4.2 (which are constructed by the NN) are different from the hypothesis class used in $L_{CA}$. To overcome this difficulty, we use the theory of VC-subgraph classes to connect the VC dimension among multiple hypothesis classes, including the class of $\psi_i$, the four classes of output functions, and $\mathcal{F}$, $\log \mathcal{G}$. Then we establish the covering number bound for the class $\mathcal{F}$ and $\log \mathcal{G}$, and finally prove Theorem 4.5. To conclude, minimizing $\mathbb{E}[K_1(X)]$ over $\hat{P}$ is equivalent to minimizing $\mathbb{E}[K(X)]$, which in turn is minimizing a tight upper bound for the coverage assessment loss. Our coverage assessment loss empirically approximates $\mathbb{E}[K_1(X)]$ well, so that its minimization can ultimately reduce the conditional coverage error.

## 5 EXPERIMENTS

**Experimental Setup.** We empirically verify the effectiveness of our proposed CaNet on both synthetic examples and benchmark regression datasets. These datasets have been widely used for the evaluation of methods in regression tasks (Hernández-Lobato & Adams, 2015; Gal & Ghahramani, 2016; Lakshminarayanan et al., 2017; Rosenfeld et al., 2018; Pearce et al., 2018; Zhu et al., 2019). In addition, we adopt the same experimental procedure in Pearce et al. (2018) for data normalization and dataset splitting. To avoid overfitting, we apply a simple network architecture with only 2 hidden layers and each hidden layer has 64 neurons. For each hidden layer, the ReLU activation function is applied to capture the non-linear features. We empirically set the ensemble number $m$ to 5, as the smallest number leading to a stable prediction results. Please refer to Appendix E for implementation details, including those for baseline algorithms.

**Evaluation Metrics.** To evaluate the conditional coverage estimation of our CaNet, we examine the quality of our Ca-Module measured by the empirical coverage error $ECE_1$ over a partition $\Delta$. $\Delta$ is constructed by equally dividing the width of $[0, 1]$ into $M$ sub-intervals. The value of $M$ depends on the size of the dataset, which is determined by the following strategy: $M = \min\{\lfloor \text{the number of data in validation}/50 \rfloor + 1, 20\}$. In addition, we also report the $IW$ and $CP$, where $IW = \frac{1}{n}\sum_{i=1}^{n}(U(x_i) - L(x_i))$ (equation 3.1) and $CP = \frac{1}{n}\sum_{i=1}^{n}k_i$, (equation 3.2) to show the effectiveness of our predictor module under the high-quality criteria.

### 5.1 CONDITIONAL COVERAGE ON SYNTHETIC EXAMPLES

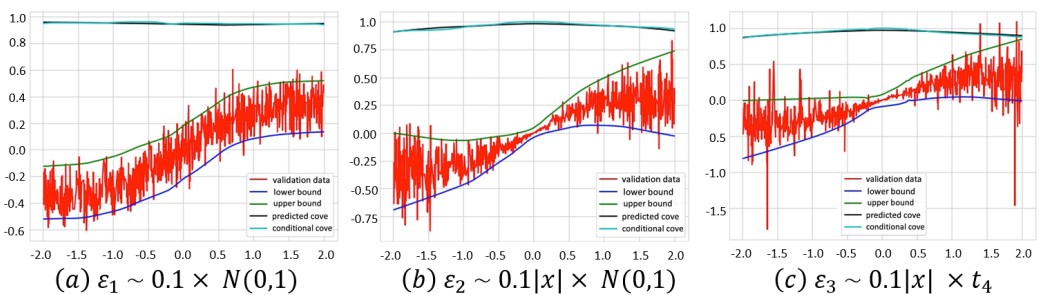

Figure 2: Prediction and conditional coverage of 95% PIs on synthetic examples. (a) $CP = 0.95$, $IW = 0.40$, $ECE_1 = 0.62\%$. (b) $CP = 0.96$, $IW = 0.40$, $ECE_1 = 0.12\%$. (c) $CP = 0.95$, $IW = 0.50$, $ECE_1 = 0.65\%$. The predicted coverage estimation from CaNet is highly consistent with the conditional coverage under different noise settings.

In this section, we conduct a series of experiments on synthetic examples to directly compare our prediction results with the ground-truth conditional coverage. In these examples, the conditional coverage can be analytically calculated under the known data distribution. Figure 2 compares the conditional coverage with our predicted coverage under the following settings: $x \sim \text{Uniform}[-2, 2]$ and $y|x$ is drawn from $f_i(x) = \frac{1}{3}\sin(x) + \varepsilon_i(x)$, $x \in [-2, 2]$ where

$$\varepsilon_1(x) = 0.1 \times N(0,1), \quad \varepsilon_2(x) = 0.1|x| \times N(0,1), \quad \varepsilon_3(x) = 0.1|x| \times t_4.$$

$N(0, 1)$ is the standard Gaussian variable and $t_4$ is the standard $t$ random variable with 4 degrees of freedom. Then, the conditional coverage in Definition 2.1 can be analytically calculated as:

$$\mathbb{P}[Y \in [L(X), U(X)]|L, U, X = x] = F_i(U(x) - \frac{1}{3}\sin(x)) - F_i(L(x) - \frac{1}{3}\sin(x))$$

| Dataset | NNKCDE | | QRF | | SCL | | QD-Ens | | CaNet: $\lambda_2 = 0.1^5$ for all datasets | | | | |
|---|---|---|---|---|---|---|---|---|---|---|---|---|---|
| | $CP$ | $IW$ | $CP$ | $IW$ | $CP$ | $IW$ | $CP$ | $IW$ | $CP$ | $IW$ | $ECE_1$ | $\lambda_1$ | $\lambda_3$ |
| Boston | 0.95 | 1.54 | 0.96 | 2.22 | 0.97 | 1.45 | 0.92 | 1.16 | 0.95 | **1.04** | 1.38% | 6.0 | 1800 |
| Concrete | 0.95 | 1.85 | 0.99 | 2.53 | 0.96 | 1.54 | 0.94 | 1.09 | 0.95 | **1.13** | 0.24% | 6.5 | 1000 |
| Energy | 0.97 | 0.54 | 0.98 | 0.87 | 0.95 | 0.77 | 0.97 | 0.47 | 0.99 | **0.37** | 0.76% | 5.0 | 500 |
| Kin8nm | 0.99 | 2.76 | 0.99 | 3.27 | 0.94 | 1.20 | 0.96 | 1.25 | 0.95 | **1.04** | 1.32% | 3.6 | 300 |
| Plant | 0.95 | 0.88 | 0.97 | 1.05 | 0.95 | 0.88 | 0.95 | 0.86 | 0.95 | **0.84** | 0.38% | 3.3 | 700 |
| Protein | 0.93 | 1.98 | 0.98 | 2.42 | 0.95 | 2.81 | 0.95 | 2.27 | 0.95 | **2.26** | 0.41% | 8.3 | 300 |
| Wine | 0.96 | 2.64 | 0.93 | 3.18 | 0.96 | 3.44 | 0.92 | 2.33 | 0.95 | **2.59** | 0.42% | 19 | 1100 |
| Yacht | 0.95 | 1.15 | 0.95 | 1.72 | 0.95 | 0.57 | 0.96 | 0.17 | 0.98 | **0.16** | 0.80% | 1.6 | 500 |

Table 1: Evaluation metrics of different models on benchmark datasets. The $CP$ values are marked in blue if they meet the 95% prediction level. The best IW results, marked in bold, are achieved by models with the smallest IW value among those that meet the 95% prediction level. Our model outperforms the baseline algorithms on high-quality PI generation. Meanwhile, it provides accurate coverage estimation on real-world datasets.

| Kin8nm: $\lambda_2 = 0.1^5, \lambda_3 = 300$ | | | | Plant: $\lambda_2 = 0.1^5, \lambda_3 = 700$ | | | | Protein: $\lambda_2 = 0.1^5, \lambda_3 = 300$ | | | |
|---|---|---|---|---|---|---|---|---|---|---|---|
| $\lambda_1$ | $CP$ | $IW$ | $ECE_1$ | $\lambda_1$ | $CP$ | $IW$ | $ECE_1$ | $\lambda_1$ | $CP$ | $IW$ | $ECE_1$ |
| 1.9 | 0.83 | 0.73 | 1.54% | 1.4 | 0.80 | 0.56 | 1.23% | 4.8 | 0.82 | 1.56 | 0.82% |
| 2.1 | 0.85 | 0.77 | 1.59% | 1.5 | 0.84 | 0.62 | 1.36% | 5.1 | 0.86 | 1.70 | 0.71% |
| 2.3 | 0.87 | 0.80 | 1.47% | 1.6 | 0.85 | 0.63 | 0.86% | 5.4 | 0.88 | 1.79 | 1.25% |
| 2.4 | 0.88 | 0.84 | 1.21% | 1.8 | 0.86 | 0.64 | 1.36% | 5.6 | 0.90 | 1.84 | 1.04% |
| 2.6 | 0.90 | 0.87 | 1.50% | 2.0 | 0.89 | 0.70 | 0.51% | 6.3 | 0.91 | 1.95 | 0.66% |
| 2.9 | 0.91 | 0.89 | 0.50% | 2.4 | 0.91 | 0.75 | 1.15% | 6.5 | 0.92 | 2.07 | 1.36% |
| 3.1 | 0.92 | 0.94 | 0.49% | 2.6 | 0.92 | 0.80 | 1.29% | 7.0 | 0.93 | 2.11 | 1.23% |
| 3.4 | 0.94 | 1.00 | 0.82% | 3.0 | 0.94 | 0.83 | 0.23% | 7.8 | 0.94 | 2.13 | 0.99% |
| 3.6 | 0.95 | 1.04 | 1.32% | 3.3 | 0.95 | 0.84 | 0.38% | 8.3 | 0.95 | 2.26 | 0.41% |
| 3.8 | 0.96 | 1.06 | 1.25% | 4.0 | 0.96 | 0.89 | 0.80% | 9.3 | 0.96 | 2.36 | 0.52% |

Table 2: $ECE_1$ results of our model on benchmark datasets with different coverage probabilities. Our CaNet achieves robust performance on real-world datasets at different prediction levels.

where $F_i$ is the cumulative distribution function of $N(0, 0.1^2)$ for $i = 1$, $N(0, (0.1x)^2)$ for $i = 2$, and $0.1|x| \times t_4$ for $i = 3$. As shown in Figure 2, the conditional coverages of high-quality PIs at different points diverge among each other and they deviate from the marginal coverage. Thus, having access to the marginal coverage for the whole dataset is not sufficient for decision making, which highlights the need of conditional coverage. In addition, the predicted coverage estimator of our model is highly consistent with the conditional coverage on all of the synthetic examples. These results confirm that our CaNet can accurately estimate the conditional coverage on noisy datasets.

## 5.2 PERFORMANCE OF PIS ON BENCHMARK DATASETS

In this section, we compare the predictor module of our CaNet on real-world benchmark datasets with following baseline algorithms: (1) nearest-neighbors kernel conditional density estimation (NNKCDE) (Dalmasso et al., 2020), (2) quantile regression forest (QRF) (Meinshausen, 2006), (3) split conformal learning (SCL) (Lei et al., 2018) and (4) the quality-driven PI method (QD-Ens) (Pearce et al., 2018). We quote the results from (Pearce et al., 2018) as a comparison since we share the same experiment setup. Table 1 reports the results of $CP$, $IW$ and $ECE_1$ for generating PIs at 95% prediction level on benchmark datasets. We employed the criteria in Pearce et al. (2018) to evaluate the performance of PIs: the best $IW$ is achieved by the model with the smallest $IW$ value among those with $CP \geq 95\%$. As can be seen, our model achieves the best performance on PI generation under the high-quality criteria. With special consideration on interval width quality, it obtains the smallest average interval width ($IW$) while maintaining high coverage probability ($CP \geq 95\%$) on all datasets.

### 5.3 PERFORMANCE OF COVERAGE ESTIMATOR ON BENCHMARK DATASETS

**Coverage for 95% PIs.** We use $ECE_1$ to evaluate the coverage estimation performance on real-world datasets as the conditional coverage is unknown. As shown in Table 1, the $ECE_1$ on all experiments are generally around or less than 1%, with better performance on larger datasets. The coverage estimators produced by CaNet have small $ECE_1$ values, which are very close to the perfect-calibrated coverage estimators (Definition 2.2). Compared with $ECE_1$ values obtained from the state-of-the-art algorithms in classification tasks (Guo et al., 2017; Kull et al., 2019), the $ECE_1$ values from CaNet are similar and sometimes less than their post-calibrated $ECE_1$ results (usually around 1% to 3%), even though the size of most regression datasets are smaller than the classification datasets. These results further demonstrate that our CaNet can accurately estimate the coverage information of 95% high-quality PIs on real-world regression tasks.

**Coverage for PIs at Different Prediction Levels.** We conduct multiple experiments on different PI prediction levels to show the robustness of our CaNet. By only modifying the parameter $\lambda_1$ in Equation 3.5, our Ca-Module could get access to different levels of coverage probability. Table 2 reports the $CP$, $IW$ and $ECE_1$ values from the CaNet at different PI prediction levels on three benchmark datasets. Results for more datasets can be found in Appendix E. As can be seen, all $ECE_1$ values in Table 2 are fairly small ($\sim 1\%$), demonstrating the stability of our proposed model. Thus, our CaNet can provide accurate coverage estimation on PIs at different prediction levels. These results demonstrate the robustness of our CaNet on real-world datasets, further suggesting its broad applicability.

## 6 RELATED WORK

**Prediction Interval Estimation.** High-quality PIs, which can be viewed via a constrained optimization problem where the constraint concerns marginal coverage and the objective is the PI width, has been extensively studied in Khosravi et al. (2010; 2011); Galván et al. (2017); Pearce et al. (2018); Rosenfeld et al. (2018); Zhang et al. (2019); Zhu et al. (2019). Such intervals are in the same spirit as the highest density intervals in statistics (Box & Tiao, 2011). While powerful, these approaches could not directly provide the conditional coverage information investigated in this work. Coverage-only criteria, on the other hand, focus solely on coverage satisfaction as the guarantee. These approaches include conformal learning (CL) and its conditional variants (Vovk et al., 2005; 2009; Lei & Wasserman, 2014; Lei et al., 2015; 2018; Kuchibhotla & Ramdas, 2019; Romano et al., 2019; Barber et al., 2019a;b). CL is desirably distribution- or model-free, and in many cases enjoys finite-sample guarantees. However, unlike high-quality PIs, they do not explicitly account for the interval width as a quality metric. We also mention conditional density estimation (Holmes et al., 2007; Dutordoir et al., 2018; Izbicki & Lee, 2016; Dalmasso et al., 2020; Freeman et al., 2017; Izbicki et al., 2017) and closely relatedly quantile regression (Koenker & Hallock, 2001; Meinshausen, 2006) as PI construction approaches by converting from the estimated conditional quantile function. These approaches focus on the quality of conditional distribution/quantile, instead of the high-quality criterion only. In this work, we use deep learning to construct high-quality PIs (Khosravi et al., 2010; Pearce et al., 2018; Kivaranovic et al., 2020).

**Uncertainty Measurement in Deep Learning.** The Bayesian framework offers principled approaches for model uncertainty measurement by computing the posterior distribution over the NN parameters (MacKay, 1992; Neal, 2012). These approaches can also be used to construct PIs. However, they provide a different perspective from the frequentist view taken in this paper, and they focus on the parameter uncertainty of the NN instead of the coverage over a test point. In addition, exact Bayesian inference is computationally intractable for deep neural networks, making it less practical to implement. Gal & Ghahramani (2016) applied a Monte Carlo dropout method to proxy the inference. Directly generated from the networks, softmax response is also commonly used for uncertainty measurement on deep learning models (Bridle, 1990; Lakshminarayanan et al., 2017; Geifman et al., 2018; Sensoy et al., 2018; Ozbulak et al., 2018). Moreover, Niculescu-Mizil & Caruana (2005) showed that NNs typically produce well-calibrated probabilities on binary classification tasks without the need of any post-hoc techniques. In this paper, we also follow this line of work and use the softmax output to access the model uncertainty information.

## 7  CONCLUSION

In this paper, we identify and investigate the conditional coverage estimation problem for high-quality PIs, which is critical for risk-based decision making in regression fields. To address the challenge, we propose an end-to-end algorithm with two powerful modules: coverage assessment module and predictor module. Benefited from these powerful modules, our model can generate and validate the coverage estimator without any requirement for further post-processing steps. In addition, we conduct theoretical analysis to show the effectiveness of our proposed model. Experimental results on synthetic examples and benchmark datasets further demonstrate that our model can robustly provide accurate coverage estimation while simultaneously producing a high-quality PI. Moreover, our Ca-Module can be easily integrated into other deep-learning-based algorithms to get access to their coverage information, opening up more opportunities for broad applications. In the future, we will extend our work by conducting comparison studies with Bayesian methods.

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

This Appendix presents further results and discussions and it consists of five parts. Appendix A gives more detailed properties on coverage estimator and coverage error. Appendix B contains the mathematical argument for Theorem 4.1. Appendix C discusses how to achieve Assumption 4.2 and the proof of Theorem 4.3 and 4.5. Appendix D presents our algorithm details as a supplement to Section 3. Appendix E illustrates experimental details and more experimental results.

# A    FURTHER DETAILS ON COVERAGE ESTIMATOR AND COVERAGE ERROR

## A.1    COVERAGE PROBABILITY TYPES OF PIs

Zhang et al. (2019) introduce the following four coverage probability types of PIs. In general, most of the coverage in PIs considered in the literature falls into one of these types.

Type I: $\mathbb{P}[Y \in [L(X), U(X)]]$ (marginal coverage);

Type II: $\mathbb{P}[Y \in [L(X), U(X)]|L, U]$ (conditional coverage given the PI);

Type III: $\mathbb{P}[Y \in [L(X), U(X)]|X = x]$ (conditional coverage given $X = x$);

Type IV: $\mathbb{P}[Y \in [L(X), U(X)]|L, U, X = x]$ (conditional coverage given the PI and $X = x$).

Note that in high-quality criteria, only Type II coverage is considered in the constraint but Type IV coverage is lacking. Since Type I and III are not considered in our paper, for simplicity Type II coverage is called the marginal coverage, and Type IV coverage is called the conditional coverage in Definition 2.1.

Throughout Appendix, we let $A(x) := \mathbb{P}[Y \in [L(X), U(X)]|L, U, X = x]$ denote the conditional coverage in Definition 2.1. Moreover, since we only concern Type II and IV coverage in this work, we make the following convention. Throughout the Appendix A-B, $\mathbb{P}$ and $\mathbb{E}$ should be understood as probability and expectation conditional on $L, U$. So for $A(x)$, we could simply write $A(x) := \mathbb{P}[Y \in [L(X), U(X)]|X = x]$ and omit "conditional on $L, U$".

## A.2    THE TERMINOLOGY "PERFECT-CALIBRATED"

The terminology "perfect-calibrated" is borrowed from the confidence calibration in classification tasks. We first review the name of "confidence" in classification.

Confidence calibration is the problem of predicting probability estimates representative of the true correctness likelihood (Guo et al., 2017). Intuitively, an reliable confidence should reflect the true correctness likelihood of the prediction (Kumar et al., 2019). For example, given 100 predictions, each with confidence of 0.8, we expect that 80 should be correctly classified (Guo et al., 2017).

Now let $h$ be the prediction of any models, which is a map from $\mathcal{X}$ to $\mathcal{Y}$, trained on the data $\mathcal{D}$. According to their definition, the "best" confidence map $\hat{P}$ should be the true probability of correctness:

$$\hat{P}(x) = \mathbb{E}[\mathbf{1}_{h(X) \text{ is correct for } Y}|h, X = x], \quad \forall x \in \mathcal{X}. \tag{A.1}$$

which is a measureable function from $\mathcal{X}$ to $[0, 1]$. In particular, we consider $h$ as a PI in regression tasks and "correctness" of a PI is naturally defined as the success of coverage on the outcome $Y$. Then the right hand side of Equation (A.1) becomes

$$\mathbb{E}[\mathbf{1}_{h(X) \text{ is correct for } Y}|h, X = x] = \mathbb{E}[\mathbf{1}_{Y \in [L(X), U(X)]}|L, U, X = x]$$
$$= \mathbb{P}[Y \in [L(X), U(X)]|L, U, X = x].$$

which is the conditional coverage in our Definition 2.1.

In addition, Guo et al. (2017) introduce the perfect-calibrated confidence as follows:

$$\mathbb{P}(\hat{Y} = Y|\hat{Y}, \hat{P} = p) = p, \quad \forall p \in [0, 1]$$

where $\hat{Y}$ is the class prediction and $\hat{Y} = Y$ means that the predicted and true class label coincide. Obviously, if $\hat{P}(X) = \mathbb{P}(\hat{Y}(X) = Y|X)$, i.e., the "best" confidence, then the above equality holds. Transferring this idea into PIs, we can naturally define the perfect-calibrated coverage estimator as

$$p = \mathbb{E}[\mathbf{1}_{h(X) \text{ is correct for } Y}|L, U, \hat{P} = p] = \mathbb{P}[Y \in [L(X), U(X)]|L, U, \hat{P} = p], \quad \forall p \in [0, 1].$$

which is the conditional coverage in our Definition 2.2.

### A.3 Details on Coverage Estimator

A perfect-calibrated coverage estimator inherits some properties of a conditional coverage. For example, both of them have the following interpretation: If we have 1000 testing points for a PI, each with the same conditional/perfect-calibrated coverage 0.9, then approximately 900 of them are correctly covered by the PI. Note that the conditional coverage is uniquely defined, but a perfect-calibrated coverage estimator is not necessarily so. Moreover, we have the following facts: (a) The conditional coverage is always perfect-calibrated, but not vice versa. (b) A perfect-calibrated coverage estimator can be viewed as an averaged conditional coverage. (c) A perfect-calibrated coverage estimator is less "informative" than the conditional coverage.

**A perfect-calibrated coverage estimator is an averaged conditional coverage.** Let $\hat{P}$ be a (general) coverage estimator. We have

$$\mathbb{P}[Y \in [L(X), U(X)] | \hat{P}(X) = \hat{P}(x)]$$
$$= \mathbb{P}[Y \in [L(X), U(X)] | X \in \hat{P}^{-1}(\hat{P}(x))]$$
$$= \frac{\mathbb{P}[Y \in [L(X), U(X)], X \in \hat{P}^{-1}(\hat{P}(x))]}{\mathbb{P}[X \in \hat{P}^{-1}(\hat{P}(x))]}$$
$$= \frac{\int_{t \in \hat{P}^{-1}(\hat{P}(x))} \mathbb{E}[\mathbf{1}_{Y \in [L(X), U(X)]} | X = t] \mathbb{P}[X \in dt]}{\mathbb{P}[X \in \hat{P}^{-1}(\hat{P}(x))]}$$
$$= \frac{\int_{t \in \hat{P}^{-1}(\hat{P}(x))} A(t) \mathbb{P}[X \in dt]}{\mathbb{P}[X \in \hat{P}^{-1}(\hat{P}(x))]}$$

Suppose $\hat{P}$ is a perfect-calibrated coverage estimator. Then we have

$$\hat{P}(x) = \frac{\int_{t \in \hat{P}^{-1}(\hat{P}(x))} A(t) \mathbb{P}[X \in dt]}{\mathbb{P}[X \in \hat{P}^{-1}(\hat{P}(x))]}$$

which implies that $\hat{P}(x)$ is a weighted average of $A(t)$ over the set $\hat{P}^{-1}(\hat{P}(x))$ with weights based on the marginal distribution of $X$.

**A conditional coverage is perfect-calibrated.** If $\hat{P}(x) = A(x)$, then $A(t) = A(x)$ for any $t \in A^{-1}(A(x))$ so

$$\frac{\int_{t \in A^{-1}(A(x))} A(t) \mathbb{P}[X \in dt]}{\mathbb{P}[X \in A^{-1}(A(x))]} = \frac{A(x) \int_{t \in A^{-1}(A(x))} \mathbb{P}[X \in dt]}{\mathbb{P}[X \in A^{-1}(A(x))]} = A(x).$$

This shows that $A(x)$ must be a perfect-calibrated coverage estimator. Another way to see this is taking conditional expectation given $A(X) = p$ in the Definition 2.1. Then we get

$$p = \mathbb{E}[A(X) | A(X) = p]$$
$$= \mathbb{E}[\mathbb{P}[Y \in [L(X), U(X)] | X] | A(X) = p]$$
$$= \mathbb{P}[Y \in [L(X), U(X)] | A(X) = p] \quad \text{by the tower property.}$$

**A perfect-calibrated coverage estimator may be less informative and may not be the conditional coverage.** Suppose we have a PI $[L(X), U(X)]$ at the exact prediction level of $1 - \alpha$, i.e., $\mathbb{P}[Y \in [L(X), U(X)]] = 1 - \alpha$. Then the constant coverage estimator

$$\hat{P}(x) = 1 - \alpha, \quad \forall x \in \mathcal{X}$$

can be viewed as an average coverage estimator over the entire space $\mathcal{X}$. It is a perfect-calibrated coverage estimator since by definition,

$$\mathbb{P}[Y \in [L(X), U(X)] | \hat{P}(X) = 1 - \alpha] = \mathbb{P}[Y \in [L(X), U(X)]] = 1 - \alpha = \hat{P}(X).$$

But it is not a conditional coverage in general (e.g., the second synthetic example in Section 5).

Now we give an extension of the definition of "perfect-calibrated" coverage estimator, allowing it to be defined on any measurable subsets.

**Definition A.1.** *A coverage estimator $\hat{P}$ is called a perfect-calibrated coverage estimator on a measurable subset $\mathcal{S} \subset \mathcal{X}$ with $\mathbb{P}(\mathcal{S}) > 0$ associated with $[L(x), U(x)]$ if it satisfies*

$$\hat{P}(x) = \mathbb{P}[Y \in [L(X), U(X)] | L, U, \hat{P}(X) = \hat{P}(x), X \in \mathcal{S}], \quad a.e. \ \hat{P}(x) \in [0, 1]. \quad \text{(A.2)}$$

*where a.e. is with respect to the probability measure on $[0, 1]$ induced by the random variable $\hat{P}(X)|_{\mathcal{S}}$. Note that the conditional probability space is standard: $(\mathcal{S}, \mathcal{F}_{\mathcal{S}} := \{A \cap \mathcal{S} : A \in \mathcal{F}\}, \mathbb{P}_{\mathcal{S}}(A \cap \mathcal{S}) := \mathbb{P}(A|\mathcal{S}))$.*

(As our convention in Section A.1, we will omit "conditional on $L, U$" for simplicity.)

**Lemma A.2.** *(a) A coverage estimator is the conditional coverage if and only if it is a perfect-calibrated coverage estimator on any positive-probability measurable subset $\mathcal{S}$ of $\mathcal{X}$.*
*(b) Suppose $\hat{P}$ is a perfect-calibrated coverage estimator on two disjoint positive-probability measurable subsets $\mathcal{S}_1, \mathcal{S}_2$. Then $\hat{P}$ is a perfect-calibrated coverage estimator on $\mathcal{S}_1 \cup \mathcal{S}_2$.*

*Proof.* (a) The proof can be found in Lemma A.4.

(b) We note that by law of total probability,

$$\begin{aligned}
&\mathbb{P}[Y \in [L(X), U(X)] | \hat{P}(X) = p, X \in \mathcal{S}_1 \cup \mathcal{S}_2] \\
=&\mathbb{P}[Y \in [L(X), U(X)] | \hat{P}(X) = p, X \in \mathcal{S}_1] \mathbb{P}[X \in \mathcal{S}_1 | \hat{P}(X) = p, X \in \mathcal{S}_1 \cup \mathcal{S}_2] \\
&+ \mathbb{P}[Y \in [L(X), U(X)] | \hat{P}(X) = p, X \in \mathcal{S}_2] \mathbb{P}[X \in \mathcal{S}_2 | \hat{P}(X) = p, X \in \mathcal{S}_1 \cup \mathcal{S}_2] \\
=&p(\mathbb{P}[X \in \mathcal{S}_1 | \hat{P}(X) = p, X \in \mathcal{S}_1 \cup \mathcal{S}_2] + \mathbb{P}[X \in \mathcal{S}_2 | \hat{P}(X) = p, X \in \mathcal{S}_1 \cup \mathcal{S}_2]) \\
=&p.
\end{aligned}$$

Hence $\hat{P}$ is a perfect-calibrated coverage estimator on $\mathcal{S}_1 \cup \mathcal{S}_2$. $\qquad\square$

Lemma A.2(a) is motivated from a theoretical point of view. It provides a guidance that in order to well resemble the conditional coverage, an estimator should be perfect-calibrated on as many subsets on the feature space as possible.

## A.4 DETAILS ON COVERAGE ERROR

In Section 2, we have introduced $CE_p$ to quantify the discrepancy between a coverage estimator and a perfect-calibrated coverage estimator, and $\widetilde{CE}_p$ to quantify the discrepancy between a coverage estimator and the conditional coverage. We note that by Hölder's inequality

$$CE_p \leq CE_q, \text{ for } 1 \leq p \leq q \leq +\infty.$$

A larger value of $p$ corresponds to a larger $CE$ value. Continuing Definition A.1, we can further introduce the calibration-based conditional coverage error on a measurable subset as follows:

**Definition A.3.** *An $L^p$ ($1 \leq p \leq +\infty$) calibration-based conditional coverage error, or coverage error for short, of a coverage estimator $\hat{P}$ on a measurable subset $\mathcal{S} \subset \mathcal{X}$ with $\mathbb{P}(\mathcal{S}) > 0$ is defined as:*

$$CE_p(\mathcal{S}) = \left\| \mathbb{P}[Y \in [L(X), U(X)] | L, U, \hat{P}(X), X \in \mathcal{S}] - \hat{P}(X) \right\|_{L^p(\mathcal{S})} \quad \text{(A.3)}$$

*where $L^p$-norm is taken with respect to the randomness of $\hat{P}(X)$ on the conditional probability space $(\mathcal{S}, \mathcal{F}_{\mathcal{S}} := \{A \cap \mathcal{S} : A \in \mathcal{F}\}, \mathbb{P}_{\mathcal{S}}(A \cap \mathcal{S}) := \mathbb{P}(A|\mathcal{S}))$. In particular, we have $CE_p := CE_p(\mathcal{X})$.*

**Lemma A.4.** *A coverage estimator $\hat{P}$ is the conditional coverage if and only if its coverage error $CE_p(\mathcal{S}) = 0$ for any measurable subset $\mathcal{S} \subset \mathcal{X}$ with $\mathbb{P}(\mathcal{S}) > 0$. In particular, a coverage estimator is the conditional coverage if and only if it is a perfect-calibrated coverage estimator on any measurable subset $\mathcal{S}$ of $\mathcal{X}$ with $\mathbb{P}(\mathcal{S}) > 0$.*

*Proof.* We first show that the conditional coverage is perfect-calibrated. Taking conditional expectation given $\{A(X) = p, X \in \mathcal{S}\}$ in the Definition 2.1. Then we get

$$
\begin{aligned}
p &= \mathbb{E}[A(X)|A(X) = p, X \in \mathcal{S}] \\
&= \mathbb{E}[\mathbb{P}[Y \in [L(X), U(X)]|X]|A(X) = p, X \in \mathcal{S}] \\
&= \mathbb{P}[Y \in [L(X), U(X)]|A(X) = p, X \in \mathcal{S}] \quad \text{by the tower property.}
\end{aligned}
$$

So $A(x)$ is a perfect-calibrated coverage estimator on any measurable subset $\mathcal{S}$ with $\mathbb{P}(\mathcal{S}) > 0$. Hence $CE_p(\mathcal{S}) = 0$ for any measurable subsets $\mathcal{S} \subset \mathcal{X}$ with $\mathbb{P}(\mathcal{S}) > 0$.

On the other hand, similarly to Section A.3, we can express

$$
\mathbb{P}[Y \in [L(X), U(X)]|\hat{P}(X) = \hat{P}(x), X \in \mathcal{S}] = \frac{\int_{t \in \hat{P}^{-1}(\hat{P}(x)) \cap \mathcal{S}} A(t)\mathbb{P}[X \in dt]}{\mathbb{P}[X \in \hat{P}^{-1}(\hat{P}(x)) \cap \mathcal{S}]}.
$$

Suppose $\hat{P}(x)$ is not the conditional coverage, then $\mathbb{P}[\hat{P}(X) \neq A(X)] > 0$. Without loss of generality, we assume $\mathbb{P}[\hat{P}(X) > A(X)] > 0$. Let $\mathcal{S}_0 := \{x \in \mathcal{X} : \hat{P}(x) > A(x)\}$. Note that $\mathcal{S}_0 = \cup_{n=1}^{+\infty} \{x \in \mathcal{X} : \hat{P}(x) > A(x) + \frac{1}{n}\}$. Since $\mathbb{P}(\mathcal{S}_0) > 0$, there exists a $n_0$ such that $\mathcal{S} := \{x \in \mathcal{X} : \hat{P}(x) > A(x) + \frac{1}{n_0}\}$ and $\mathbb{P}(\mathcal{S}) > 0$.

Then for $x \in \mathcal{S}$, we have

$$
\frac{\int_{t \in \hat{P}^{-1}(\hat{P}(x)) \cap \mathcal{S}} A(t)\mathbb{P}[X \in dt]}{\mathbb{P}[X \in \hat{P}^{-1}(\hat{P}(x)) \cap \mathcal{S}]} \leq \frac{\int_{t \in \hat{P}^{-1}(\hat{P}(x)) \cap \mathcal{S}} (\hat{P}(t) - \frac{1}{n_0})\mathbb{P}[X \in dt]}{\mathbb{P}[X \in \hat{P}^{-1}(\hat{P}(x)) \cap \mathcal{S}]} = \hat{P}(x) - \frac{1}{n_0}.
$$

Then we have

$$
CE_p(\mathcal{S}) \geq CE_1(\mathcal{S}) \geq \frac{1}{n_0} > 0
$$

so $CE_p(\mathcal{S}) > 0$, which is a contradiction. Hence $\hat{P}(x)$ is the conditional coverage. $\square$

Next, we describe in detail how to discretize the right-hand-side of equation 2.4 for empirical calculation. We construct a discrete version of (2.2) and then introduce an empirical counterpart of $CE_p$ (2.3), which we refer to as $L^p$ empirical calibration-based conditional coverage error $ECE_p$. The ideas behind these are natural extensions of the classification case (Guo et al., 2017; Kull et al., 2019; Kumar et al., 2019; Nixon et al., 2019) into PIs.

We consider the following partition $\Delta$ of $[0, 1]$. Let $[0, 1]$ be divided into $M$ intervals $I_m = (a_{m-1}, a_m]$ ($m = 1, \cdots, M$) where $0 = a_0 \leq a_1 \leq \cdots \leq a_M = 1$. Let $B_m = \{i = 1, \cdots, n : \hat{P}(x_i) \in I_m\}$, i.e., the set (bin) of indices $i$ of samples whose prediction coverage estimator $\hat{P}(x_i)$ falls into the interval $I_m$. Note that coverage estimators that are close to each other will fall into the same interval. The coverage probability (i.e., the proportion of successful coverage) in $B_m$ is defined as:

$$
\text{cp}(B_m) = \frac{1}{|B_m|} \sum_{i \in B_m} \mathbf{1}_{y_i \in [L(x_i), U(x_i)]}. \tag{A.4}
$$

The average estimated coverage in $B_m$ is defined as:

$$
\text{cove}(B_m) = \frac{1}{|B_m|} \sum_{i \in B_m} \hat{P}(x_i) \tag{A.5}
$$

where $\hat{P}(x_i)$ is the coverage estimator for sample $i$. $\text{cp}(B_m)$ and $\text{cove}(B_m)$ approximate the left and right hand sides of (2.2) respectively in the interval $I_m$. A perfect-calibrated coverage estimator should satisfy $\text{cp}(B_m) = \text{cove}(B_m)$ for all $m \in \{1, \cdots, M\}$. The diagram of $\text{cp}(B_m)$ versus $\text{cove}(B_m)$ for all $m \in \{1, \cdots, M\}$ is called the reliability diagram in some literature (Guo et al., 2017).

Based on the partition $\Delta$, we can introduce an empirical version of $CE_p$ (which we refer to as $ECE_p$) as:

**Definition A.5.** *The $L^p$ empirical calibration-based conditional coverage error ($ECE_p$) of a coverage estimator $\hat{P}$ is defined as*

$$ECE_p = \left\| (cp(B_{m(i)}) - cove(B_{m(i)}))_{i=1,2,\cdots,n} \right\|_{l^p}. \tag{A.6}$$

*where the $l^p$ is the standard p-norm in $\mathbb{R}^n$ and $B_{m(i)}$ is the bin containing sample i.*

Equivalently,

$$ECE_p = \left( \sum_{m=1}^{M} \frac{|B_m|}{n} |cp(B_m) - cove(B_m)|^p \right)^{\frac{1}{p}}, \quad 1 \le p < +\infty.$$

$$ECE_\infty = \max_{m=1,2,\cdots,M} |cp(B_m) - cove(B_m)|.$$

Note that $ECE_p$ is discontinuous and cannot be used easily for training with gradient-based methods. Therefore, we use the coverage assessment loss $L_{CA}$ introduced in Section 3 for conditional coverage estimation.

Finally, we give some explanations about why the conditional coverage error $\widetilde{CE}_p$ cannot be used easily for training. Unlike $L_{CA}$ which is unbiased and provides guaranteed estimation accuracy for $\mathbb{E}[K_1(X)]$ (Theorem 4.5), it is in general not easy to establish an empirical calculation for $\widetilde{CE}_p$. Take $\widetilde{CE}_2$ as an instance. A heuristic argument is to use

$$\hat{\theta} := \frac{1}{n} \sum_{i=1}^{n} (k_i - \hat{P}(x_i))^2$$

to approximate

$$\widetilde{CE}_2^2 = \mathbb{E}[|A(X) - \hat{P}(X)|^2]$$

where $k_i = 0$ or $1$ indicates whether each data point has been captured by the PIs; see Section 3. Unfortunately, $\hat{\theta}$ is in general not an unbiased estimator of $\widetilde{CE}_2^2$ due to the following observations:

$$
\begin{aligned}
\mathbb{E}[\hat{\theta}] &= \mathbb{E}[(k_1 - \hat{P}(x_1))^2] \\
&= \mathbb{E}\Big[ \mathbb{E}[(k_1 - \hat{P}(x_1))^2 | x_1] \Big] \\
&= \mathbb{E}\Big[ \mathbb{E}[k_1^2 | x_1] - 2\hat{P}(x_1)\mathbb{E}[k_1|x_1] + \hat{P}(x_1)^2 \Big] \\
&= \mathbb{E}\Big[ \mathbb{E}[k_1 | x_1] - 2\hat{P}(x_1)\mathbb{E}[k_1|x_1] + \hat{P}(x_1)^2 \Big] \\
&= \mathbb{E}\Big[ A(x_1) - 2\hat{P}(x_1)A(x_1) + \hat{P}(x_1)^2 \Big] \\
&= \mathbb{E}\Big[ A(x_1)^2 - 2\hat{P}(x_1)A(x_1) + \hat{P}(x_1)^2 \Big] + \mathbb{E}[A(x_1) - A(x_1)^2] \\
&= \mathbb{E}[|A(X) - \hat{P}(X)|^2] + \mathbb{E}[A(X) - A(X)^2].
\end{aligned}
$$

Hence $\mathbb{E}[\hat{\theta}] \ne \widetilde{CE}_2^2$ unless $A(X) = 0$ or $1$ almost surely. Note that the gap $\mathbb{E}[A(X) - A(X)^2]$ does not depend on $n$ and thus not vanish as $n$ grows. To obtain a reasonable estimator for $\widetilde{CE}_p$, one need more information about $A(x)$ besides the marginal coverage $\mathbb{E}[A(X)]$. However, this information is usually hard to obtain locally because of the nature of (possibly high-dimensional) feature space. In fact, this has led us to the idea of "dimension reduction" of the sample space to the space $[0, 1]$ and the definition of $CE_p$.

# B   MATHEMATICAL DEVELOPMENTS FOR THEOREM 4.1

This section proves Theorem 4.1: we show that both coverage error and conditional coverage error are tightly bounded above by the expectation of a Kullback–Leibler divergence-type random variable $K_1(x)$. This means that minimizing $\mathbb{E}[K_1(X)]$ can recover the true conditional coverage and effectively reduce the coverage error. The proof consists of several inequalities regarding coverage errors and their relations. We first begin with the following connection between $CE_p$ and $\widetilde{CE}_p$.

**Theorem B.1.** *For any PI and its associated $\hat{P}$, the $L^p$ coverage error is always less than or equal to the $L^p$ conditional coverage error, i.e.,*

$$CE_p \leq \widetilde{CE}_p, \quad \forall 1 \leq p \leq +\infty$$

*Proof.* We note that the function $t \mapsto |t|^p$ is a convex function. We also note that $\sigma(\hat{P}(X)) \subset \sigma(X)$ where $\sigma(Y)$ represents the $\sigma$-field generated by a random variable $Y$.

$$
\begin{aligned}
\widetilde{CE}_p^p &= \mathbb{E}\left[\left|A(X) - \hat{P}(X)\right|^p\right] \\
&= \mathbb{E}\left[\mathbb{E}|A(x) - \hat{P}(X)|^p|\hat{P}(X)]\right] \\
&\geq \mathbb{E}\left[\left|\mathbb{E}[A(x) - \hat{P}(X)|\hat{P}(X)]\right|^p\right] \quad \text{by Jensen's inequality} \\
&= \mathbb{E}\left[\left|\mathbb{E}[\mathbf{1}_{Y \in [L(X), U(X)]}|\hat{P}(X)] - \hat{P}(X)\right|^p\right] \quad \text{by the tower property} \\
&= CE_p^p
\end{aligned}
$$

Therefore we have

$$CE_p \leq \widetilde{CE}_p, \quad \forall 1 \leq p \leq +\infty.$$

$\square$

Next we have the following bounds on $L^p$ conditional coverage error:

**Theorem B.2.** *The $L^p$ conditional coverage error is bounded above by a power function of the $L^2$ conditional coverage error. Formally,*

$$\widetilde{CE}_p \leq \widetilde{CE}_2^{\alpha_p}, \quad \forall 1 \leq p \leq +\infty,$$

*where $\alpha_p = 1, \forall 1 \leq p \leq 2$ and $\alpha_p = \frac{2}{p}, \forall 2 \leq p \leq +\infty$.*

*Proof.* By Hölder's inequality,

$$\widetilde{CE}_p \leq \widetilde{CE}_2, \text{ if } 1 \leq p \leq 2.$$

Since

$$0 \leq |A(x) - \hat{P}(x)| \leq 1,$$

then,

$$|A(x) - \hat{P}(x)|^p \leq |A(x) - \hat{P}(x)|^2, \quad \forall p \geq 2$$

and thus

$$\widetilde{CE}_p^p = \mathbb{E}[|A(X) - \hat{P}(X)|^p] \leq \mathbb{E}[|A(X) - \hat{P}(X)|^2] \leq \widetilde{CE}_2^2, \quad \forall p \geq 2.$$

$\square$

Next, recall that

$$K(x) = A(x) \log\left(\frac{A(x)}{\hat{P}(x)}\right) + (1 - A(x)) \log\left(\frac{1 - A(x)}{1 - \hat{P}(x)}\right),$$

$$K_0(x) = A(x) \log(A(x)) + (1 - A(x)) \log(1 - A(x)),$$

$$K_1(x) = -A(x) \log(\hat{P}(x)) - (1 - A(x)) \log(1 - \hat{P}(x)).$$

**Theorem B.3.** *The $L^2$ conditional coverage error is bounded above by the expectation of $K(x)$. Formally,*

$$\widetilde{CE}_2^{\alpha_p} \leq \left(\frac{1}{2}\mathbb{E}[K(X)]\right)^{\alpha_p/2},$$

*where $\alpha_p$ is defined in Theorem B.2.*

*Proof.* For any fixed $x$, consider two random variables with Bernoulli distributions:

$$W_1 = \begin{cases} 1 & \text{w.p. } A(x), \\ 0 & \text{w.p. } 1 - A(x). \end{cases}$$

$$W_2 = \begin{cases} 1 & \text{w.p. } \hat{P}(x), \\ 0 & \text{w.p. } 1 - \hat{P}(x). \end{cases}$$

Let $P_i$ be the distribution of $W_i$. It follows from Pinsker's inequality, e.g., Theorem 2.16 in (Massart, 2007), that

$$\|P_1 - P_2\|_{TV}^2 \leq \frac{1}{2} K(P_1, P_2).$$

where $TV$ denotes the total variation distance and $K$ denotes the KL divergence. Since $P_i$ is the Bernoulli distribution, we can express it as

$$|A(x) - P(x)|^2 \leq \frac{1}{2} K(x)$$

Taking expectation, we obtain

$$\mathbb{E}[|A(X) - P(X)|^2] \leq \frac{1}{2}\mathbb{E}[K(X)].$$

Hence,

$$\widetilde{CE}_2^{\alpha_p} \leq \left(\frac{1}{2}\mathbb{E}[K(X)]\right)^{\alpha_p/2}.$$

$\square$

Combining Theorem B.1, B.2 and B.3, we immediately conclude that:

**Theorem B.4** (Restated Theorem 4.1)**.**

$$CE_p \leq \widetilde{CE}_p \leq \widetilde{CE}_2^{\alpha_p} \leq \left(\frac{1}{2}\mathbb{E}[K(X)]\right)^{\alpha_p/2}, \quad \forall 1 \leq p \leq +\infty$$

*where $\alpha_p$ is defined in Theorem B.2. Moreover, all inequalities are attainable, e.g., if $\hat{P}(x)$ is the conditional coverage $A(x)$.*

## C  JUSTIFICATION OF ASSUMPTION 4.2 AND MATHEMATICAL DEVELOPMENTS FOR THEOREM 4.5

In this section, we analyze the rationality of Assumption 4.2 and build essential ingredients for proving Theorem 4.3 and 4.5. The difficulty in analyzing Theorem 4.5 lies in the fact that the hypothesis classes in Assumption 4.2 (which are constructed by the NN) are different from the hypothesis class used in $L_{CA}$. To overcome this difficulty, we use the theory of VC-subgraph classes to analyze the connection between the VC dimension of the two hypothesis classes.

### C.1  REVIEW OF THE VC DIMENSION

For self-contained purpose, we first review the definition of the VC-subgraph class and VC dimension.

**Definition C.1.** *Consider an arbitrary collection $\{x_1, \cdots, x_n\}$ of points in a set $\mathcal{X}$ and a collection $\mathcal{C}$ of subsets of $\mathcal{X}$. We say that $\mathcal{C}$ shatters $\{x_1, \cdots, x_n\}$ if all of $2^n$ possible subsets of $\{x_1, \cdots, x_n\}$ can be written as $A = C \cap \{x_1, \cdots, x_n\}$ for some $C \in \mathcal{C}$. The VC dimension $V(\mathcal{C})$ of the class $\mathcal{C}$ is the smallest $n$ for which no set of size $n$ $\{x_1, \cdots, x_n\}$ is shattered by $\mathcal{C}$. If $\mathcal{C}$ shatters sets of arbitrarily large size, we set $V(\mathcal{C}) = \infty$. We say that $\mathcal{C}$ is a VC-class if $V(\mathcal{C}) < \infty$.*

In some literature, the VC dimension $V(\mathcal{C})$ of the class $\mathcal{C}$ is alternatively defined as the largest $n$ for which there exists a set of size $n$ $\{x_1, \cdots, x_n\}$ shattered by $\mathcal{C}$, i.e., it is the value in definition C.1 minus 1. We can more formally define the VC dimension by the growth function as follows:

**Definition C.2.** *Define the $n^{th}$ shatter coefficient (or growth function) of $\mathcal{C}$ as*

$$\Pi_{\mathcal{C}}(n) := \max_{x_1, \cdots, x_n} |\{A : A = C \cap \{x_1, \cdots, x_n\} \text{ for some } C \in \mathcal{C}\}|$$

*Then*

$$V(\mathcal{C}) := \inf\{n : \Pi_{\mathcal{C}}(n) < 2^n\}.$$

**Definition C.3.** *For a function $f : \mathcal{X} \to \mathbb{R}$, the subset of $\mathcal{X} \times \mathbb{R}$ given by $\{(x, t) : t < f(x)\}$ is the (open) subgraph of $f$. A collection $\mathcal{F}$ of measurable real functions on the sample space $\mathcal{X}$ is a VC-subgraph class or VC-class, if the collection of all subgraphs of functions in $\mathcal{F}$ forms a VC-class of sets (as sets in $\mathcal{X} \times \mathbb{R}$). Let $V(\mathcal{F})$ denote the VC dimension of the set of subgraphs of $\mathcal{F}$.*

**Lemma C.4.** *In Definition C.3, the open subgraph of $f$, $\{(x, t) : t < f(x)\}$, can be replaced by the close subgraph $\{(x, t) : t \leq f(x)\}$, the close supergraph $\{(x, t) : t \geq f(x)\}$ or the open supergraph $\{(x, t) : t > f(x)\}$. All of them lead to the equivalent definition of the VC-class and the equal VC dimension.*

*Proof.* This result follows from Lemma 9.33 and Lemma 9.9(iv) in Kosorok (2007). □

For indicator functions of sets, we have the following equivalence.

**Lemma C.5.** *For any class $\mathcal{C}$ of sets in a set $\mathcal{X}$, the class $\mathcal{F}_{\mathcal{C}}$ of indicator functions of sets in $\mathcal{C}$ is a VC-class if and only if $\mathcal{C}$ is a VC-class. Moreover, whenever at least one of $\mathcal{C}$ or $\mathcal{F}_{\mathcal{C}}$ is VC-class, the respective VC dimensions are equal.*

*Proof.* This is Lemma 9.8 in Kosorok (2007). Note that the sets of $\mathcal{C}$ are in $\mathcal{X}$ while the subgraphs of functions of $\mathcal{F}_{\mathcal{C}}$ are in $\mathcal{X} \times \mathbb{R}$. □

## C.2 JUSTIFYING ASSUMPTION 4.2

We first restate the assumption:

**Assumption C.6** (Restated Assumption 4.2). *The four classes of functions $([L(x), U(x)], \hat{P}(x), 1 - \hat{P}(x))$ output by the neural network (NN) in Figure 1 have finite VC dimensions, say they are bounded above by $V_0$.*

In Figure 1, the output four neurons of the NN are denoted as $(L(x), U(x), \hat{P}(x), 1 - \hat{P}(x))$. We further let $(\psi_1(x), \psi_2(x), \psi_3(x), \psi_4(x))$ denote the pre-activated values of $(L(x), U(x), \hat{P}(x), 1 - \hat{P}(x))$. In other words,

$$L(x) = \min(\psi_1(x), \psi_2(x)),$$
$$U(x) = \max(\psi_1(x), \psi_2(x)),$$
$$\hat{P}(x) = \text{softmax}(\psi_3(x) || \psi_4(x)) = \sigma(\psi_3(x) - \psi_4(x)),$$
$$1 - \hat{P}(x) = \text{softmax}(\psi_4(x) || \psi_3(x)) = \sigma(\psi_4(x) - \psi_3(x)),$$

where $\sigma$ is the sigmoid function. Let the function classes

$$\mathcal{H}_1 = \{L(x) : L \text{ is output by the the NN}\},$$
$$\mathcal{H}_2 = \{U(x) : U \text{ is output by the the NN}\},$$
$$\mathcal{G} = \{\hat{P}(x) : \hat{P} \text{ is output by the NN}\},$$
$$1 - \mathcal{G} = \{1 - \hat{P}(x) : \hat{P} \text{ is output by the NN}\}.$$

Assumption 4.2 holds for a wide range of NNs, in particular the one we adopt in the experiments (where we use the ReLU-activated NN to construct $\psi_i, i = 1, 2, 3, 4$; see Section 5). Our first result is to concretely show that the four NN outputs above, $\mathcal{H}_1, \mathcal{H}_2, \mathcal{G}$ and $1 - \mathcal{G}$, under the ReLU setting, all have finite VC dimensions and thus satisfy Assumption 4.2.

**Theorem C.7** (Restated Theorem 4.3). *Suppose $\psi_i, i = 1, 2, 3, 4$ are the pre-activated output neurons of the NN in Figure 1 using the ReLU activation function. Then Assumption 4.2 holds. Moreover, suppose the NN has $W$ parameters and $U$ computation units (nodes). Then $V_0 = O(WU)$.*

*Proof.* First, we look at $L(x)$ and $U(x)$. Note that the class of $\psi_i$ ($i = 1, 2$) is constructed by a NN with the ReLU activation function. Therefore by Theorem 8 in Bartlett et al. (2019) (see also C.8 below),

$$V(\{\psi_1\}) = O(WU) < \infty, \quad V(\{\psi_2\}) = O(WU) < \infty.$$

By Lemma 9.9 (i) in Kosorok (2007), we have

$$V(\mathcal{H}_1) = V(\{\min(\psi_1, \psi_2)\}) \leq V(\{\psi_1\}) + V(\{\psi_2\}) - 1 = O(WU) < \infty.$$

By Lemma 9.9 (ii) in Kosorok (2007), we have

$$V(\mathcal{H}_2) = V(\{\max(\psi_1, \psi_2)\}) \leq V(\{\psi_1\}) + V(\{\psi_2\}) - 1 = O(WU) < \infty.$$

Next, we look at $\hat{P}(x)$ and $1 - \hat{P}(x)$. We add an additional neuron after the layer where $\psi_3$, $\psi_4$ stand. This neuron is defined as $\psi_5 = \psi_3 - \psi_4$ which is a linear combination of $\psi_3$ and $\psi_4$. Note that the class of $\psi_5$ is constructed by a NN with the ReLU activation function and linear activation function (by adding one unit and two parameters in the originial NN). Therefore by Theorem 8 in Bartlett et al. (2019),

$$V(\{\psi_5\}) = O(WU) < \infty$$

By Lemma 9.9 (viii) in Kosorok (2007), we have

$$V(\mathcal{G}) = V(\{\sigma(\psi_5)\}) \leq V(\{\psi_5\}) = O(WU) < \infty.$$

since $\sigma$ is a monotone function. Again, by Lemma 9.9 (viii) in Kosorok (2007), we have

$$V(1 - \mathcal{G}) \leq V(\mathcal{G}) = O(WU) < \infty.$$

since $t \mapsto 1 - t$ is a monotone function. $\qquad\square$

We also list some results for other activations here. From these results, and using the same argument as above, we see that Assumption 4.2 holds similarly for all these activations.

**Lemma C.8.** *Suppose the class of functions is constructed by a NN with $W$ parameters and $U$ units with activation functions that are piecewise polynomials with at most $p$ pieces and of degree at most $d$. Then it has VC dimension $O(WU \log((d + 1)p))$.*

*Proof.* This is Theorem 8 in Bartlett et al. (2019). $\qquad\square$

Note that the activation functions in Lemma C.8 include in particular the ReLU activation and linear activation.

**Lemma C.9.** *Suppose the class of functions is constructed by a NN with $W$ parameters with binary as well as linear activation function. Then it has VC dimension $O(W^2)$.*

*Proof.* This is Theorem 5 in Sontag (1998). $\qquad\square$

**Lemma C.10.** *Suppose the class of functions is constructed by a NN with $W$ parameters and $U$ units with activation function that is the standard sigmoid function (except that the output unit being a linear threshold unit). Then it has VC dimension $O(W^2 U^2)$.*

*Proof.* This is Theorem 8.13 in Anthony & Bartlett (2009). $\qquad\square$

C.3 CONNECTIONS AMONG DIFFERENT HYPOTHESIS CLASSES

To prove Theorem 4.5, we need to study several building blocks on the relations between different hypothesis classes. Our first observation is:

**Theorem C.11.** *Suppose $V(\mathcal{G}) < +\infty$. Then all of the following classes have VC dimension $\leq V(\mathcal{G})$:*

$$1 - \mathcal{G} := \{1 - \hat{P}(x) : \hat{P} \text{ is output by the NN}\}.$$

$$\mathcal{G}' := \log(\mathcal{G}) := \{\log(\hat{P}(x)) : \hat{P} \text{ is output by the NN}\}.$$

$$\log(1 - \mathcal{G}) := \{\log(1 - \hat{P}(x)) : \hat{P} \text{ is output by the NN}\}.$$

*Proof.* The result follows from Lemma 9.9 (viii) in Kosorok (2007) since all of the transformations are monotone functions. □

Our second observation is about

$$\mathcal{F} = \{f(x,y) = I_{y \in [L(x), U(x)]} : L, U \text{ are output by the NN}\}.$$

Note that the domain of functions in $\mathcal{F}$ is different from the domain of functions in $\mathcal{H}_i$ ($i = 1, 2$) as it includes the outcome space. Below we derive a result that connects the VC dimension of $\mathcal{H}_i$ with that of $\mathcal{F}$.

**Theorem C.12.** *Suppose $V(\mathcal{H}_i) \leq V_0$ ($i = 1, 2$). We have that*

$$V(1 - \mathcal{F}) \leq V(\mathcal{F}) \leq 10(V_0 - 1) < +\infty$$

*where*

$$1 - \mathcal{F} := \{1 - f(x,y) : f \in \mathcal{F}\}.$$

*Proof.* The first inequality follows from Lemma 9.9 (viii) in Kosorok (2007). We consider the following two classes:

$$\mathcal{F}_1 := \{I_{L(x) \leq t} : L \in \mathcal{H}_1, t \in \mathbb{R}\},$$

$$\mathcal{F}_2 := \{I_{U(x) \geq t} : U \in \mathcal{H}_2, t \in \mathbb{R}\}.$$

Since the functions in $\mathcal{F}_1$ are all indicator functions, by Lemma C.5,

$$V(\mathcal{F}_1) = V(\{\{(x,t) : L(x) \leq t\} : L \in \mathcal{H}_1, t \in \mathbb{R}\}).$$

Note that the latter is the VC dimension of the close supergraphs of all functions in $\mathcal{H}_1$. Then by Definition C.3 and Lemma C.4, we have

$$V(\{\{(x,t) : L(x) \leq t\} : L \in \mathcal{H}_1, t \in \mathbb{R}\}) = V(\mathcal{H}_1)$$

Therefore we have

$$V(\mathcal{F}_1) = V(\mathcal{H}_1) \leq V_0$$

Similarly,

$$V(\mathcal{F}_2) = V(\mathcal{H}_2) \leq V_0$$

Note that we can write

$$I_{y \in [L(x), U(x)]} = I_{L(x) \leq y} I_{U(x) \geq y}$$

By the definition of growth functions,

$$\begin{aligned}
\Pi_{\mathcal{F}}(m) &:= \max_{(x_1, y_1), \cdots, (x_m, y_m)} \left| \{(I_{y_1 \in [L(x_1), U(x_1)]}, \cdots, I_{y_m \in [L(x_m), U(x_m)]}) : L \in \mathcal{H}_1, U \in \mathcal{H}_2\} \right| \\
&\leq \max_{(x_1, y_1), \cdots, (x_m, y_m)} \left| \{(I_{L(x_1) \leq y_1}, \cdots, I_{L(x_m) \leq y_m}) : L \in \mathcal{H}_1\} \right| \times \\
&\quad \max_{(x_1, y_1), \cdots, (x_m, y_m)} \left| \{(I_{U(x_1) \geq y_1}, \cdots, I_{U(x_m) \geq y_m}) : U \in \mathcal{H}_2\} \right| \\
&= \Pi_{\mathcal{F}_1}(m) \Pi_{\mathcal{F}_2}(m) \\
&\leq \left( \frac{em}{V_0 - 1} \right)^{2(V_0 - 1)}
\end{aligned}$$

for all $m \geq V_0$ where the last inequality is due to the Sauer–Shelah lemma. Taking $m = 10(V_0 - 1)$, we obtain

$$\left( \frac{em}{V_0 - 1} \right)^{2(V_0 - 1)} = (10e)^{2(V_0 - 1)} \leq 750^{V_0 - 1} < 2^m$$

Combining the above inequality, we have

$$\Pi_{\mathcal{F}}(m) < 2^m.$$

This shows that $V(\mathcal{F}) \leq m = 10(V_0 - 1)$. □

### C.4 PROOF OF THEOREM 4.5

This subsection proves Theorem 4.5. Recall that

$$\mathcal{F} = \{I_{y \in [L(x), U(x)]} : L, U \text{ are output by the NN}\},$$

$$\mathcal{G} = \{\hat{P}(x) : \hat{P} \text{ is output by the NN}\}.$$

$$\mathcal{G}' := \log(\mathcal{G}) := \{\log(\hat{P}(x)) : \hat{P} \text{ is output by the NN}\}.$$

Let $N(\epsilon, \mathcal{F}, L^2(Q))$ denote the covering number, i.e., the minimal number of balls $\{g : \|g - h\|_{L^2(Q)} < \epsilon\}$ of radius $\epsilon$ needed to cover the set $\mathcal{F}$. We need the following bounds:

**Lemma C.13.** *Suppose $\mathcal{F}$ is a class of functions $f : \mathcal{X} \times \mathcal{Y} \to [0, 1]$ with a finite VC dimension $V(\mathcal{F})$. For every $0 < \epsilon < 1$,*

$$\sup_Q \log N(\epsilon, \mathcal{F}, L^2(Q)) \leq K_2 \left(\frac{1}{\epsilon}\right)^{\frac{1}{e}}$$

*where the constant $K_2$ depends on $V(\mathcal{F})$ only.*

*Proof.* It follows from Theorem 2.6.7 in Van der Vaart & Wellner (1996) that there exists a universal constant $K$ such that

$$\sup_Q N(\epsilon, \mathcal{F}, L^2(Q)) \leq KV(\mathcal{F})(16e)^{V(\mathcal{F})} \left(\frac{1}{\epsilon}\right)^{V(\mathcal{F})-1}$$

for any $0 < \epsilon < 1$. Since

$$\left(\frac{1}{\epsilon}\right)^{\frac{1}{e}} \geq \max\left(\log\left(\frac{1}{\epsilon}\right), 1\right), \quad \forall 0 < \epsilon < 1,$$

we have

$$\sup_Q \log N(\epsilon, \mathcal{F}, L^2(Q)) \leq K_3 + (V(\mathcal{F}) - 1) \log\left(\frac{1}{\epsilon}\right) \leq K_2 \left(\frac{1}{\epsilon}\right)^{\frac{1}{e}}$$

where $K_3 := \log(KV(\mathcal{F})(16e)^{V(\mathcal{F})})$ and $K_2 = K_3 + V(\mathcal{F}) - 1$ only depending on $V(\mathcal{F})$. $\square$

We remark that a similar result can also be obtained for the class $1 - \mathcal{F}$ by Theorem C.12.

**Lemma C.14.** *Suppose $\mathcal{G}$ is a class of functions $\hat{P} : \mathcal{X} \to [0, 1]$ with a finite VC dimension $V(\mathcal{G})$ and $|\log(\hat{P}(x))| \leq M$. Let $\mathcal{G}' := \{\log(\hat{P}) : \hat{P} \in \mathcal{G}\}$. Then, for every $0 < \epsilon < 1$,*

$$\sup_Q \log N(\epsilon M, \mathcal{G}', L^2(Q)) \leq K_2 \left(\frac{1}{\epsilon}\right)^{\frac{1}{e}}$$

*where the constant $K_2$ depends on $V(\mathcal{G})$ only.*

*Proof.* First note that $\phi(t) := \log(t)$ is a monotone function. Hence $\mathcal{G}' := \{\log(\hat{P}) : \hat{P} \in \mathcal{G}\}$ is a VC-class with VC dimension $\leq V(\mathcal{G})$ by Lemma 9.9 (viii) in Kosorok (2007). The rest of the proof is similar to C.13. $\square$

We remark that a similar result can also be obtained for the class $\log(1 - \mathcal{G})$ by Theorem C.11.

Next we restate Assumption 4.4:

**Assumption C.15** (Restated Assumption 4.4). $|\log(\hat{P}(x))| \leq M$, $|\log(1 - \hat{P}(x))| \leq M$ *for all $x$ and $\hat{P}$.*

As discussed in Section 4, this is a natural assumption in practice because $\log(\hat{P}(x))$ and $\log(1 - \hat{P}(x))$ are replaced by $\log(\hat{P}(x) + \epsilon)$ and $\log(1 - \hat{P}(x) + \epsilon)$ respectively to avoid explosion when implementing the algorithm. In particular, in our experiments in Section 5, $\epsilon = 0.1^6$ and thus $M = 14$.

We are now ready to prove Theorem 4.5:

**Theorem C.16** (Restated Theorem 4.5). *Suppose Assumptions 4.2 and 4.4 hold. The training data* $\mathcal{D} = \{(x_i, y_i), \ i = 1, 2, \cdots, n\}$ *where* $(x_i, y_i)$ *are i.i.d. samples* $\sim \pi$. *Recall that the (hard) coverage estimator assessment loss is*

$$L_{CA} = -\frac{1}{n}\sum_{i=1}^{n}\Big(f(x_i, y_i)\log(\hat{P}(x_i)) + (1 - f(x_i, y_i))\log(1 - \hat{P}(x_i))\Big).$$

*Then for any* $t > 0$, *we have*

$$\mathbb{P}\left(\sup_{f \in \mathcal{F}, \hat{P} \in \mathcal{G}} |L_{CA} - \mathbb{E}[K_1(X)]| \geq t\right) \leq C^* e^{-\frac{nt^2}{16M^2}}$$

*where* $C^*$ *only depends on* $V_0$ *in Assumption 4.2.*

*Proof.* Note that $\mathbb{E}[f(x_i, y_i)|x_i] = A(x_i)$ for any fixed $L$ and $U$. Taking expectation on $L_{CA}$, we have

$$\begin{aligned}
\mathbb{E}[L_{CA}] &= \mathbb{E}\left[\mathbb{E}[L_{CA}|x_1, x_2, \cdots, x_n]\right] \\
&= \mathbb{E}\left[-\frac{1}{n}\sum_{i=1}^{n}\Big(A(x_i)\log(\hat{P}(x_i)) + (1 - A(x_i))\log(1 - \hat{P}(x_i))\Big)\right] \\
&= \mathbb{E}[K_1(X)].
\end{aligned}$$

We consider the first part $A(x)\log(\hat{P}(x))$. The second part can be done using the same argument. Note that by Theorem 9.15 in Kosorok (2007), we have

$$\sup_Q \log N(\epsilon M, \mathcal{F} \cdot \mathcal{G}', L^2(Q)) \leq \sup_Q \log N(\epsilon/2, \mathcal{F}, L^2(Q)) + \sup_Q \log N(\epsilon M/2, \mathcal{G}', L^2(Q)).$$

Consider the class $\frac{1}{2} + \frac{1}{2M}\mathcal{F} \cdot \mathcal{G}' := \{\frac{1}{2} + \frac{1}{2M}(f(x, y)\log(\hat{P}(x))) : f \in \mathcal{F}, \log(\hat{P}) \in \mathcal{G}'\}$ which consists of functions taking values in $[0, 1]$. We have

$$\begin{aligned}
&\sup_Q \log N(\epsilon, \frac{1}{2} + \frac{1}{2M}\mathcal{F} \cdot \mathcal{G}', L^2(Q)) \\
&= \sup_Q \log N(2\epsilon M, \mathcal{F} \cdot \mathcal{G}', L^2(Q)) \\
&\leq \sup_Q \log N(\epsilon, \mathcal{F}, L^2(Q)) + \sup_Q \log N(\epsilon M, \mathcal{G}', L^2(Q)) \\
&\leq K_2\left(\frac{1}{\epsilon}\right)^{1/e}
\end{aligned}$$

where the last inequality follows from Lemma C.13 and Lemma C.14, and $K_2$ only depends on $V(\mathcal{F})$ and $V(\mathcal{G})$. (Recall that we have shown $V(\mathcal{G}') \leq V(\mathcal{G})$ in Lemma C.14.) Moreover, by Theorems C.11 and C.12, we can claim that $K_2$ only depends on $V_0$. This inequality shows that $\frac{1}{2} + \frac{1}{2M}\mathcal{F} \cdot \mathcal{G}'$ satisfies the conditions in Theorem 2.14.10 in Van der Vaart & Wellner (1996) and thus for every $\delta > 0$ and $t > 0$,

$$\mathbb{P}\left(\sup_{\phi \in \frac{1}{2} + \frac{1}{2M}\mathcal{F} \cdot \mathcal{G}'}\left|\frac{1}{n}\sum_{i=1}^{n}\phi(x_i, y_i) - \mathbb{E}[\phi(x, y)]\right| \geq t\right) \leq Ce^{D(\sqrt{n}t)^{U+\delta}}e^{-2nt^2}$$

where $U = \frac{\frac{1}{e}(6 - \frac{1}{e})}{2 + \frac{1}{e}} < 1$ and the constants $C$ and $D$ depend on $K_2$ and $\delta$ only. Let $\delta = 1 - U$. Note that

$$-2(\sqrt{n}t)^2 + D(\sqrt{n}t) \leq -(\sqrt{n}t)^2 + (D/2)^2.$$

Hence we have

$$\mathbb{P}\left(\sup_{\phi \in \frac{1}{2} + \frac{1}{2M}\mathcal{F} \cdot \mathcal{G}'}\left|\frac{1}{n}\sum_{i=1}^{n}\phi(x_i, y_i) - \mathbb{E}[\phi(x, y)]\right| \geq t\right) \leq C^* e^{-nt^2}$$

where $C^*$ only depends on $K_2$, or, only depends on $V_0$.

This shows that

$$\mathbb{P}\left(\sup_{f\in\mathcal{F},\hat{P}\in\mathcal{G}}\left|\frac{1}{n}\sum_{i=1}^{n}f(x_i,y_i)\log(\hat{P}(x_i))-\mathbb{E}[A(x)\log(\hat{P}(x))]\right|\geq t\right)\leq C^*e^{-\frac{nt^2}{4M^2}}.$$

A similar result can be established for the second part since the hypothesis classes there have been studied in Theorem C.11 and C.12:

$$\mathbb{P}\left(\sup_{f\in\mathcal{F},\hat{P}\in\mathcal{G}}\left|\frac{1}{n}\sum_{i=1}^{n}(1-f(x_i,y_i))\log(1-\hat{P}(x_i))-\mathbb{E}[(1-A(x))\log(1-\hat{P}(x))]\right|\geq t\right)$$

$$\leq C^*e^{-\frac{nt^2}{4M^2}}.$$

Combining the two parts and noting the following fact:

$$\{\sup|\gamma+\beta|\geq t\}$$
$$\subset\{\sup|\gamma|+\sup|\beta|\geq t\}$$
$$\subset\{\sup|\gamma|\geq\frac{t}{2}\}\cup\{\sup|\beta|\geq\frac{t}{2}\},$$

we conclude that

$$\mathbb{P}\left(\sup_{f\in\mathcal{F},\hat{P}\in\mathcal{G}}|L_{CA}-\mathbb{E}[K_1(x)]|\geq t\right)\leq C^*e^{-\frac{nt^2}{16M^2}}$$

where $C^*$ only depends on $V_0$. $\qquad\square$

Lastly, the following corollary explicitly connects our theoretical developments to the experimental setup:

**Corollary C.17.** *Suppose the NN is designed as the one specified in the experiments (Section 5). The training data $\mathcal{D}=\{(x_i,y_i),\ i=1,2,\cdots,n\}$ where $(x_i,y_i)$ are i.i.d. samples $\sim\pi$. Then for any $t>0$, we have*

$$\mathbb{P}\left(\sup_{f\in\mathcal{F},\hat{P}\in\mathcal{G}}|L_{CA}-\mathbb{E}[K_1(X)]|\geq t\right)\leq C^*e^{-\frac{nt^2}{16M^2}}.$$

*where $C^*$ only depends on $V_0$ in Assumption 4.2.*

*Proof.* We note that Assumptions 4.2 and 4.4 hold in this case by Theorem 4.3 and the observation after Assumption 4.4. So Theorem 4.5 implies Corollary C.17. $\qquad\square$

## D ALGORITHM DETAILS

We provide additional algorithm details for our framework in Section 3. Algorithm 1 is the description of our tuning procedure for hyper-parameters $\lambda_1, \lambda_2, \lambda_3$. Let the marginal coverage probability $CP_{\mathcal{D}'}$ and the average coverage estimation $AC_{\mathcal{D}'}$ on the validation set $\mathcal{D}'$ be defined as

$$CP_{\mathcal{D}'}=\frac{1}{|\mathcal{D}'|}\sum_{i\in\mathcal{D}'}\mathbf{1}_{y_i\in[L(x_i),U(x_i)]},\quad AC_{\mathcal{D}'}=\frac{1}{|\mathcal{D}'|}\sum_{i\in\mathcal{D}'}\hat{P}(x_i).$$

where $([L(x),U(x)],\hat{P}(x))$ are prediction results from the deep ensemble. Then, $\lambda_i, i=1,2,3$ are adjusted to ensure that $CP_{\mathcal{D}'}$ coincides roughly with $AC_{\mathcal{D}'}$, and $CP_{\mathcal{D}'}$ attains the target prediction level.

## E EXPERIMENTAL DETAILS AND MORE RESULTS

This section illustrates experimental details and more experimental results from our proposed model.

---

**Algorithm 1:** Tuning algorithm

---

**Goal**: Tune hyperparameters $\lambda_1$, $\lambda_2$, and $\lambda_3$;
**Input**: Prediction level $1 - \alpha$, training dataset $\mathcal{D}$, validation dataset $\mathcal{D}'$;
**Procedure**: (1) Initialize $\lambda_i$ ($i = 1, 2, 3$) so that $CP_{\mathcal{D}'}$ is nontrivial, i.e., not (almost) 0 or 1.
(2) While $CP_{\mathcal{D}'}$ is nontrivial: tune $\lambda_2$ and $\lambda_3$ so that $|CP_{\mathcal{D}'} - AC_{\mathcal{D}'}| \leq \epsilon$ (e.g., $\epsilon = 1\%$.)
(3) Otherwise tune $\lambda_1$ such that $CP_{\mathcal{D}'}$ is nontrivial. Do step 2 again until we find $\lambda_2$ and $\lambda_3$.
(4) Tune $\lambda_1$ such that $CP_{\mathcal{D}'} > 1 - \alpha$ where $\lambda_2$ and $\lambda_3$ are fixed from (3).
**Output**: $\lambda_1$, $\lambda_2$, and $\lambda_3$.

---

### E.1 EXPERIMENTAL DETAILS

Table 3 gives a detailed description about the datasets we use. These open-access real-world benchmark regression datasets are widely used for the evaluation of methods in regression tasks (Hernández-Lobato & Adams, 2015; Gal & Ghahramani, 2016; Lakshminarayanan et al., 2017; Rosenfeld et al., 2018; Pearce et al., 2018; Zhu et al., 2019).

For synthetic datasets, 2000 i.i.d data are generated for each synthetic setting and randomly split into 1000 training data and 1000 testing data. For benchmark datasets, we first do the data normalization and then randomly split 80% data for training and 20% for testing. The choice of 80%/20% split, compared to the 90%/10% split in Pearce et al. (2018), is motivated from the need to increase the test size in order to get a meaningful $ECE$ evaluation. The latter is due to that evaluating $ECE$ requires binning, where using a larger number of bins approximates more closely $CE$, but also requires a larger test size to sustain enough statistical quality for the resulting ECE estimate. This delicate tradeoff motivates us to increase the share of the test set in our split. Following Pearce et al. (2018), our hyper-parameters are selected using the validation set from a random split. Then, they are fixed during the evaluation on other random splits.

As specified in Section 2 (Equation (2.4)) and Appendix A.4 (Equation (A.6)), ECE is evaluated based on dividing [0,1] into M sub-intervals. The larger M is, the more precise is in using ECE to approximate CE, the latter being the ideal conditional coverage error estimator. On the other hand, a larger test set size is needed to support the use of a larger M without deteriorating the statistical quality of the ECE. This delicate tradeoff motivates us to increase the share of the test set in our split.

For synthetic datasets, the hyperparameters and corresponding results in Figure 2 are:
(a) $\lambda_1 = 1.7, \lambda_2 = 10^{-5}, \lambda_3 = 1500, CP = 0.95, IW = 0.40, ECE_1 = 0.62\%$.
(b) $\lambda_1 = 1.9, \lambda_2 = 10^{-5}, \lambda_3 = 1000, CP = 0.96, IW = 0.40, ECE_1 = 0.12\%$.
(c) $\lambda_1 = 3.4, \lambda_2 = 10^{-5}, \lambda_3 = 1000, CP = 0.95, IW = 0.50, ECE_1 = 0.65\%$.

For benchmark datasets, the implementation details for baseline algorithms in Table 1 are:

(1) Nearest-neighbors kernel conditional density estimation (NNKCDE). The algorithm is based on Section 2.1 in Dalmasso et al. (2020). We use the same Python code provided by Dalmasso et al. (2020) with the default Gaussian kernel. Two tuning parameters, i.e., the number of nearest neighbors $k$ and the bandwidth $h$ of the smoothing kernel, are chosen in a principled way by minimizing the CDE loss on validation data, the same way as in Dalmasso et al. (2020).

(2) Quantile regression forest (QRF). The algorithm is based on Meinshausen (2006). We use the *RandomForestQuantileRegressor* from the package *scikit-garden* in Python.

(3) Split conformal learning (SCL). The algorithm based on Algorithm 2 in Lei et al. (2018). The regression algorithm inside SCL that we use is a neural network with mean square loss. The neural network has the same structure of hidden layers as in Section 5.

### E.2 ADDITIONAL EXPERIMENTAL RESULTS

Table 4 gives additional experimental results.

| Dataset | N | d | Open-access Link |
|---|---|---|---|
| Boston: Boston Housing | 506 | 13 | kaggle.com/c/boston-housing |
| Concrete: Concrete Strength | 1030 | 8 | kaggle.com/aakashphadtare/concrete-data |
| Energy: Energy Efficiency | 768 | 8 | kaggle.com/elikplim/eergy-efficiency-dataset |
| Kin8nm | 8192 | 8 | openml.org/d/189 |
| Plant: Combined Cycle Power Plant | 9568 | 4 | kaggle.com/gova26/airpressure |
| Protein: Protein Structure | 45730 | 9 | networkrepository.com/CASP.php |
| Wine: Red Wine Quality | 1599 | 11 | kaggle.com/uciml/red-wine-quality-cortez-et-al-2009 |
| Yacht: Yacht Hydrodynamics | 308 | 6 | archive.ics.uci.edu/ml/datasets/yacht+hydrodynamics |

Table 3: Full names and details of benchmarking regression datasets. $N$ is the number of samples in the dataset and $d$ is the dimension of the feature vector.

| Dataset | $\lambda_3$ | 1-st experiment | | | | 2-nd experiment | | | | 3-rd experiment | | | |
|---|---|---|---|---|---|---|---|---|---|---|---|---|---|
| | | $\lambda_1$ | $CP$ | $IW$ | $ECE_1$ | $\lambda_1$ | $CP$ | $IW$ | $ECE_1$ | $\lambda_1$ | $CP$ | $IW$ | $ECE_1$ |
| Boston | 1800 | 3.5 | 0.87 | 0.72 | 1.25% | 4.5 | 0.89 | 0.85 | 0.96% | 6.0 | 0.95 | 1.04 | 1.38% |
| Concrete | 1000 | 3.0 | 0.87 | 0.88 | 1.03% | 5.0 | 0.93 | 1.07 | 1.03% | 6.5 | 0.95 | 1.13 | 0.24% |
| Kin8nm | 300 | 2.1 | 0.85 | 0.77 | 1.59% | 2.9 | 0.91 | 0.89 | 0.50% | 3.6 | 0.95 | 1.04 | 1.32% |
| Plant | 700 | 1.6 | 0.85 | 0.63 | 0.86% | 2.4 | 0.91 | 0.75 | 1.15% | 3.3 | 0.95 | 0.85 | 0.38% |
| Protein | 300 | 5.1 | 0.86 | 1.70 | 0.71% | 6.3 | 0.91 | 1.95 | 0.66% | 8.3 | 0.95 | 2.26 | 0.41% |
| Wine | 1100 | 9.0 | 0.85 | 1.59 | 0.98% | 15 | 0.91 | 2.14 | 1.28% | 19 | 0.95 | 2.59 | 0.42% |
| Yacht | 500 | 1.4 | 0.94 | 0.13 | 0.13% | 1.5 | 0.96 | 0.14 | 1.51% | 1.6 | 0.98 | 0.16 | 0.80% |
| Synthetic1 | 1500 | 1.0 | 0.89 | 0.34 | 0.73% | 1.3 | 0.93 | 0.37 | 0.55% | 1.7 | 0.95 | 0.40 | 0.62% |
| Synthetic2 | 1000 | 1.1 | 0.89 | 0.31 | 1.01% | 1.4 | 0.92 | 0.34 | 1.27% | 1.9 | 0.96 | 0.40 | 0.12% |
| Synthetic3 | 1000 | 1.3 | 0.87 | 0.34 | 0.79% | 2.4 | 0.91 | 0.42 | 0.82% | 3.4 | 0.95 | 0.50 | 0.65% |

Table 4: Evaluation metrics of our CaNet on benchmark datasets and synthetic examples with different coverage probabilities.

### E.3 COMPARISONS WITH A TWO-STAGE APPROACH

We compare the performance of CaNet with a two-stage approach to further demonstrate the effectiveness of our Ca-Module. The two-stage approach is implemented with two separate steps: (1) given a regression dataset, we train a neural network to generate the prediction interval, (2) after getting the predictor, we train another network to estimate the conditional coverage of the PI from the previous stage using $L_{CA}$ loss.

Figure 3 compares the reliability diagrams (introduced in A.4) and the coverage histograms of CaNet and the two-stage approach on the dataset "Protein". The coverage histograms demonstrate the percentage of samples in each bin $B_m$ (equation A.5) for $m \in \{1, \cdots, M\}$. The average estimated coverage of our model closely matches its coverage probability, while the average estimated coverage of the two-stage algorithm is substantially lower than its coverage probability. In addition, the $ECE_1$ from CaNet (0.77%) is much lower than the $ECE_1$ from the two-stage approach (3.9%).

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

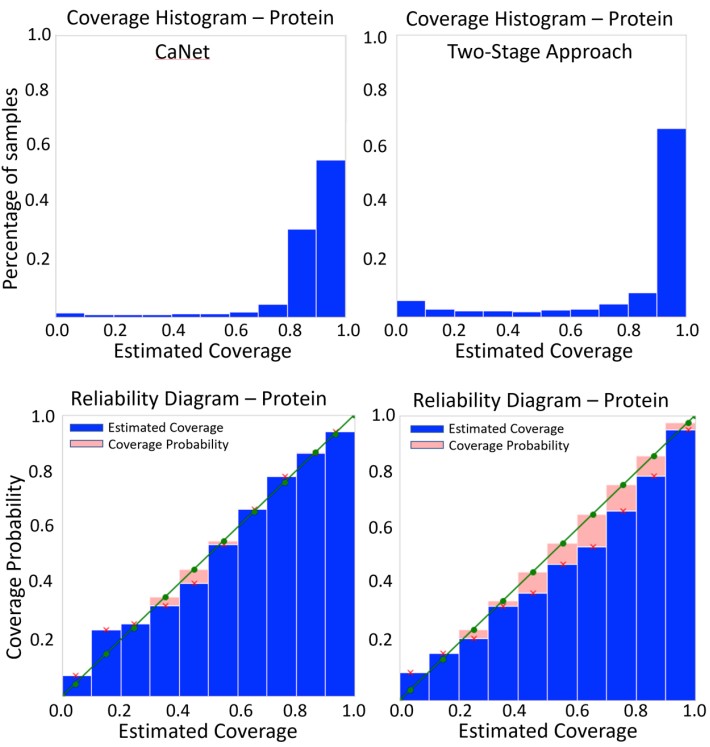

Figure 3: Coverage histograms (top) and reliability diagrams (bottom) on the dataset "Protein". Left: Our proposed algorithm ($ECE_1 = 0.77\%$, $M = 10$); Right: The two-stage algorithm ($ECE_1 = 3.9\%$, $M = 10$).

John S Bridle. Probabilistic interpretation of feedforward classification network outputs, with relationships to statistical pattern recognition. In *Neurocomputing*, pp. 227–236. Springer, 1990.

Niccolò Dalmasso, Taylor Pospisil, Ann B Lee, Rafael Izbicki, Peter E Freeman, and Alex I Malz. Conditional density estimation tools in python and r with applications to photometric redshifts and likelihood-free cosmological inference. *Astronomy and Computing*, 30:100362, 2020.

Vincent Dutordoir, Hugh Salimbeni, James Hensman, and Marc Deisenroth. Gaussian process conditional density estimation. In *Advances in neural information processing systems*, pp. 2385–2395, 2018.

Stanislav Fort, Huiyi Hu, and Balaji Lakshminarayanan. Deep ensembles: A loss landscape perspective. *arXiv preprint arXiv:1912.02757*, 2019.

Peter E Freeman, Rafael Izbicki, and Ann B Lee. A unified framework for constructing, tuning and assessing photometric redshift density estimates in a selection bias setting. *Monthly Notices of the Royal Astronomical Society*, 468(4):4556–4565, 2017.

Yarin Gal and Zoubin Ghahramani. Dropout as a bayesian approximation: Representing model uncertainty in deep learning. In *international conference on machine learning*, pp. 1050–1059, 2016.

Inés M Galván, José M Valls, Alejandro Cervantes, and Ricardo Aler. Multi-objective evolutionary optimization of prediction intervals for solar energy forecasting with neural networks. *Information Sciences*, 418:363–382, 2017.

Yonatan Geifman, Guy Uziel, and Ran El-Yaniv. Bias-reduced uncertainty estimation for deep neural classifiers. In *International Conference on Learning Representations*, 2018.

Chuan Guo, Geoff Pleiss, Yu Sun, and Kilian Q Weinberger. On calibration of modern neural networks. In *Proceedings of the 34th International Conference on Machine Learning-Volume 70*, pp. 1321–1330. JMLR. org, 2017.

Fredrik K Gustafsson, Martin Danelljan, and Thomas B Schon. Evaluating scalable bayesian deep learning methods for robust computer vision. In *Proceedings of the IEEE/CVF Conference on Computer Vision and Pattern Recognition Workshops*, pp. 318–319, 2020.

José Miguel Hernández-Lobato and Ryan Adams. Probabilistic backpropagation for scalable learning of bayesian neural networks. In *International Conference on Machine Learning*, pp. 1861–1869, 2015.

Michael P Holmes, Alexander G Gray, and Charles Lee Isbell Jr. Fast nonparametric conditional density estimation. In *Proceedings of the Twenty-Third Conference on Uncertainty in Artificial Intelligence*, pp. 175–182, 2007.

Rafael Izbicki and Ann B Lee. Nonparametric conditional density estimation in a high-dimensional regression setting. *Journal of Computational and Graphical Statistics*, 25(4):1297–1316, 2016.

Rafael Izbicki, Ann B. Lee, and Peter E. Freeman. Photo-$z$ estimation: An example of nonparametric conditional density estimation under selection bias. *Ann. Appl. Stat.*, 11(2):698–724, 06 2017. doi: 10.1214/16-AOAS1013. URL `https://doi.org/10.1214/16-AOAS1013`.

Abbas Khosravi, Saeid Nahavandi, Doug Creighton, and Amir F Atiya. Lower upper bound estimation method for construction of neural network-based prediction intervals. *IEEE transactions on neural networks*, 22(3):337–346, 2010.

Abbas Khosravi, Saeid Nahavandi, Doug Creighton, and Amir F Atiya. Comprehensive review of neural network-based prediction intervals and new advances. *IEEE Transactions on neural networks*, 22(9):1341–1356, 2011.

Danijel Kivaranovic, Kory D Johnson, and Hannes Leeb. Adaptive, distribution-free prediction intervals for deep networks. In *International Conference on Artificial Intelligence and Statistics*, pp. 4346–4356, 2020.

Roger Koenker and Kevin F Hallock. Quantile regression. *Journal of economic perspectives*, 15(4): 143–156, 2001.

Michael R Kosorok. *Introduction to empirical processes and semiparametric inference*. Springer Science & Business Media, 2007.

Arun K Kuchibhotla and Aaditya K Ramdas. Nested conformal prediction and the generalized jackknife+. *arXiv preprint arXiv:1910.10562*, 2019.

Meelis Kull, Miquel Perello Nieto, Markus Kängsepp, Telmo Silva Filho, Hao Song, and Peter Flach. Beyond temperature scaling: Obtaining well-calibrated multi-class probabilities with dirichlet calibration. In *Advances in Neural Information Processing Systems*, pp. 12295–12305, 2019.

Ananya Kumar, Percy S Liang, and Tengyu Ma. Verified uncertainty calibration. In *Advances in Neural Information Processing Systems*, pp. 3787–3798, 2019.

Balaji Lakshminarayanan, Alexander Pritzel, and Charles Blundell. Simple and scalable predictive uncertainty estimation using deep ensembles. In *Advances in Neural Information Processing Systems*, pp. 6402–6413, 2017.

Stefan Lee, Senthil Purushwalkam, Michael Cogswell, David Crandall, and Dhruv Batra. Why m heads are better than one: Training a diverse ensemble of deep networks. *arXiv preprint arXiv:1511.06314*, 2015.

Jing Lei and Larry Wasserman. Distribution-free prediction bands for non-parametric regression. *Journal of the Royal Statistical Society: Series B (Statistical Methodology)*, 76(1):71–96, 2014.

Jing Lei, Alessandro Rinaldo, and Larry Wasserman. A conformal prediction approach to explore functional data. *Annals of Mathematics and Artificial Intelligence*, 74(1-2):29–43, 2015.

Jing Lei, Max G'Sell, Alessandro Rinaldo, Ryan J Tibshirani, and Larry Wasserman. Distribution-free predictive inference for regression. *Journal of the American Statistical Association*, 113 (523):1094–1111, 2018.

David JC MacKay. *Bayesian methods for adaptive models*. PhD thesis, California Institute of Technology, 1992.

Pascal Massart. *Concentration inequalities and model selection*, volume 6. Springer, 2007.

Nicolai Meinshausen. Quantile regression forests. *Journal of Machine Learning Research*, 7(Jun): 983–999, 2006.

Radford M Neal. *Bayesian learning for neural networks*, volume 118. Springer Science & Business Media, 2012.

Alexandru Niculescu-Mizil and Rich Caruana. Predicting good probabilities with supervised learning. In *Proceedings of the 22nd International Conference on Machine Learning*, pp. 625–632, 2005.

Jeremy Nixon, Mike Dusenberry, Linchuan Zhang, Ghassen Jerfel, and Dustin Tran. Measuring calibration in deep learning. *arXiv preprint arXiv:1904.01685*, 2019.

Yaniv Ovadia, Emily Fertig, Jie Ren, Zachary Nado, David Sculley, Sebastian Nowozin, Joshua Dillon, Balaji Lakshminarayanan, and Jasper Snoek. Can you trust your model's uncertainty? evaluating predictive uncertainty under dataset shift. In *Advances in Neural Information Processing Systems*, pp. 13991–14002, 2019.

Utku Ozbulak, Wesley De Neve, and Arnout Van Messem. How the softmax output is misleading for evaluating the strength of adversarial examples. *arXiv preprint arXiv:1811.08577*, 2018.

Tim Pearce, Mohamed Zaki, Alexandra Brintrup, and Andy Neely. High-quality prediction intervals for deep learning: A distribution-free, ensembled approach. *arXiv preprint arXiv:1802.07167*, 2018.

Tim Pearce, Felix Leibfried, and Alexandra Brintrup. Uncertainty in neural networks: Approximately bayesian ensembling. In *International conference on artificial intelligence and statistics*, pp. 234–244. PMLR, 2020.

Yaniv Romano, Evan Patterson, and Emmanuel Candes. Conformalized quantile regression. In *Advances in Neural Information Processing Systems*, pp. 3543–3553, 2019.

Nir Rosenfeld, Yishay Mansour, and Elad Yom-Tov. Discriminative learning of prediction intervals. In *International Conference on Artificial Intelligence and Statistics*, pp. 347–355, 2018.

Murat Sensoy, Lance Kaplan, and Melih Kandemir. Evidential deep learning to quantify classification uncertainty. In *Advances in Neural Information Processing Systems*, pp. 3179–3189, 2018.

Eduardo D Sontag. VC dimension of neural networks. *NATO ASI Series F Computer and Systems Sciences*, 168:69–96, 1998.

Natasa Tagasovska and David Lopez-Paz. Frequentist uncertainty estimates for deep learning. *arXiv preprint arXiv:1811.00908*, 2018.

Natasa Tagasovska and David Lopez-Paz. Single-model uncertainties for deep learning. In *Advances in Neural Information Processing Systems*, pp. 6417–6428, 2019.

Aad W Van der Vaart and Jon A Wellner. *Weak Convergence and Empirical Processes with Applications to Statistics*. Springer, 1996.

Vladimir Vovk, Alex Gammerman, and Glenn Shafer. *Algorithmic learning in a random world*. Springer Science & Business Media, 2005.

Vladimir Vovk, Ilia Nouretdinov, Alex Gammerman, et al. On-line predictive linear regression. *The Annals of Statistics*, 37(3):1566–1590, 2009.

Bin Wang, Jie Lu, Zheng Yan, Huaishao Luo, Tianrui Li, Yu Zheng, and Guangquan Zhang. Deep uncertainty quantification: A machine learning approach for weather forecasting. In *Proceedings of the 25th ACM SIGKDD International Conference on Knowledge Discovery and Data Mining*, pp. 2087–2095, 2019.

Haozhe Zhang, Joshua Zimmerman, Dan Nettleton, and Daniel J Nordman. Random forest prediction intervals. *The American Statistician*, pp. 1–15, 2019.

Lin Zhu, Jiaxing Lu, and Yihong Chen. HDI-forest: highest density interval regression forest. In *Proceedings of the 28th International Joint Conference on Artificial Intelligence*, pp. 4468–4474. AAAI Press, 2019.

