# OpenReview forum: "Conditional Coverage Estimation for High-quality Prediction Intervals"
_ICLR.cc/2021/Conference — Reject_

### Official Review · AnonReviewer4 · 2020-10-18
**A novel approach for high-quality prediction intervals**

**Rating:** 8
**Confidence:** 4

**Review:**

Summary:

In the submitted paper, the authors study high-quality prediction intervals (PIs). The paper proposes a novel design of loss functions to generate PIs and conditional coverage estimates. The theoretical justification for using the conditional coverage error (in Ca-module) is presented and the numerical experiments with promising results are provided on multiple benchmark datasets.

Pros:

- The high-quality and reliable PI becomes more critical than ever as machine learning models have been used in the real-world decision-making process. This paper considers this important topic and provides a simple yet principled solution.
- The paper is well organized and theoretical results are well explained.
- Numerical experiment results on multiple synthetic and real datasets justify the practical advantages of the proposed algorithm.

Suggestion:

- Although the Bayesian framework focuses on the parameter uncertainty, as the authors mentioned in Section 6, it can be applied to generate the PIs. (Note that the posterior predictive distribution can be directly derived from the posterior distribution). A comparison study with Bayesian methods will help readers understand the advantages (or disadvantages) of the proposed method.


I vote for acceptance.

---

> ### Author Response · Authors · 2020-11-19
> **Response to Reviewer #4**
>
> We thank the reviewer for the valuable feedback and the positive opinion of our work, which is greatly encouraging for us. We agree with the reviewer that Bayesian methods are alternative methods to construct PIs, in addition to the baseline methods that we consider in this paper. We are happy to follow the proposed suggestion and consider it as our future direction, and we have added the following in the paper:
>
> In Section 6, we have added explicitly on Bayesian approaches that:
> “These approaches can also be used to construct PIs.”
>
> and at the end of Section 7:
> “In the future, we will extend our work by conducting comparison studies with Bayesian methods.”

---

### Official Review · AnonReviewer1 · 2020-10-28
**An interesting problem, but the solution does not convince**

**Rating:** 4
**Confidence:** 3

**Review:**

## Summary
The paper proposes a new framework that computes "high-quality" prediction intervals (PIs) _and_ an estimate of their conditional coverage. The latter may be regarded as an analogue of [1] for PI estimation. A theoretical justification of the loss is given under some regulatory conditions.

The problem of conditional coverage estimation is certainly well-motivated. However, there are some potential issues / questions that I would like to see clarified / answered.

---
## Strengths
1. The idea of estimating conditional coverage is an interesting one. Many methods with strong finite-sample performance are only capable of offering coverage guarantees that hold marginally. A method that is able to estimate conditional coverage with high accuracy has a potential to be useful as a diagnostic tool.

## Weaknesses / Questions
1. It isn't clear to me at all how to set $\lambda_1$ to achieve a desired coverage level. It would appear that the $L_{CP}$ component of the loss merely tracks the proportion of covered / uncovered points, so that a larger value of $\lambda_1$ is associated with more coverage, and vice versa. What is unclear to me is whether the $\lambda_1 = \lambda_1(\alpha)$ that would achieve a fixed target marginal confidence level $1-\alpha$ is known or have to be estimated via some sort of a tuning procedure. If the latter, doesn't it make the method prone to overfitting? Also, doesn't it rather invalidate the experimental results, as the comparison methods use a pre-specified target level of $1-\alpha$?
2. Is it necessary to estimate the PI and the conditional coverage simultaneously? The form of the total loss in Eq. (3.4) implies a potential tradeoff between obtaining a good PI and a good estimate of conditional coverage, but I do not see why the two objectives need to compete. On a related note, in the PI estimation problem, isn't it more interesting to estimate conditional coverage _conditional_ on a particular output of $L$ and $U$, and therefore, estimate $\hat P$ _after_ obtaining $L$ and $U$? After all, the target $A$ is already defined conditional on $L$ and $U$.

---
## Recommendation
The problem of estimating conditional coverage is an interesting one. The proposed method is not a convincing solution to the proposed problem.

---
## Additional Feedback
1. The version of the split conformal learning (SCL) implemented in experiments is somewhat outdated. [2,3] are rather more current, and produces PIs with adaptive widths, which can lead to narrower intervals in certain situations.
2. On a related note, in comparing methods that produce PIs with adaptive widths, I am not sure if the average width is interesting as a performance metric. For instance, if I somehow had access to the conditional coverage $A$ and the conditional distribution $Y | X$, I would want to compare to the width of the shortest interval with the conditional coverage. Of course, this information is unknown, but this at least suggests that the average width may be too crude.
3. Is there a typo in Eq. (3.3)? Also, the abuse of notation later in Theorem 4.5 is slightly confusing.
4. There is a typo in the line immediately above Assumption 4.2 on p. 5: $L_{CA}$ approximate -> approximate**s**
5. How difficult is it to tune the hyper-parameters? How sensitive is the method to the hyper-parameter choice?

---
## References
1. Chuan Guo, Geoff Pleiss, Yu Sun, and Kilian Q. Weinberger. On calibration of modern neural networks. ICML 2017.
2. Yaniv Romano, Evan Patterson, and Emmanuel Candes. Conformalized quantile regression. NeurIPS 2019.
3. Daniel Kivaranovic, Kory D Johnson, and Hannes Leeb. Adaptive, distribution-free prediction intervals for deep networks. AISTATS 2020.

---
## Update
I have read the revision and the rebuttal. I have also re-read the initial submission for comparison.

In the revised version, the authors have added "(4) Tune $\lambda_1$ such that $CP_{\mathcal{D}'} > 1-\alpha$  where $\lambda_2$ and $\lambda_3$ are fixed from (3)." after (3) in Algorithm 4, which substantiates their claim about the marginal coverage guarantee. As my other questions under #1 were all in response to the apparent absence of a valid calibration procedure, with the introduction of this line in the revised version, I have no further complaints about the correctness of the procedure itself. I still strongly recommend including Algorithm 1 in the main part of the paper, as a prediction interval is rather meaningless unless the associated coverage level is also known.

The biggest reason why I am keeping my score as is that after going through all the reviewing material, some of the recurring questions appear to be pointing at a larger issue with the submission.

1. It is repeatedly emphasized that the proposed method "outperforms the state-of-the-art algorithms on high-quality PI generation." This is great, except that it is hard to see *what* about the method is causing this improvement in performance. Is it the $L_{CA}$ component? Is it some non-obvious differences in architecture or in hyper-parameter tuning? Why should there be such a difference in practical performance for the simultaneous training vs a "decoupled" approach, leaving aside the practical concerns such as the computational cost?

Now that I have been thinking about this paper for awhile, I suspect that a great deal of the questions that the other reviewers and I have been asking are really about this need for *some* explanation for the improved performance. In my opinion, the current version does not provide enough evidence to *convince* the readers that the excellent empirical performance reported in Section 5 is an inevitable consequence of their novel method. This makes me cautious.

2. Throughout the review process, I couldn't escape the sense that the authors themselves have not settled on the central message. On this point, I am with R3. There is a lack of clear messaging on whether the focus is on (a) high-quality PI generation or on (b) estimating conditional coverage or on (c) both. About 3/4 of the way into the paper in my initial reading, I received the impression that the paper was definitely about (b). However, I revised my opinion and switched to (a) after going through the experimental section. After reading the first batch of the comments posted by the authors, I thought that the paper must have been about (b) all along. The last comment posted by the authors threw me into doubt yet again, however, as it seemed to indicate (c) as the correct conclusion.

In my opinion, both these issues need to be addressed before this otherwise interesting paper can be ready for publication.

---

> ### Author Response · Authors · 2020-11-19
> **Response to Reviewer #1 (2/2): Additional Feedback**
>
> **Additional Feedback**
>
> **- 1. “The version of the split conformal learning (SCL) is outdated. [2,3] are rather more current.”**
>
> We greatly thank the reviewer for pointing out these relevant works. We have incorporated these works and also some of their follow-up works in Section 6. Nonetheless, we also point out that the main contribution of this paper is to address the lack of conditional coverage consideration in these previous works. We hope the overall picture of our contribution is clear and we believe is unlikely to be changed by adding more baselines for PI generation.
>
> **- 2. “On a related note, in comparing methods that produce PIs with adaptive widths, I am not sure if the average width is interesting as a performance metric.”**
>
> The average/marginal coverage and average interval width are the most widely-used evaluation metrics [Khosravi et al. (2010; 2011); Galvan et al. (2017); Pearce et al. (2018); Rosenfeld et al. (2018); Zhang et al. (2019); Zhu et al. (2019)]. We appreciate the suggestion about using “localized” interval width as an evaluation metric. To the best of our knowledge, our paper makes the first step to consider “localized” coverage information. We believe it would be a worthwhile future direction to consider “localized” width information, though adding it into the current paper seems to disperse our main goal. Moreover, as pointed out by the reviewer, “this information is unknown” and thus it is not readily available for us to use.
>
> **- 3. “Is there a typo in Eq. (3.3)? Also, the abuse of notation later in Theorem 4.5 is slightly confusing.”**
>
> Sorry we do not find the typo suggested by the reviewer. We have nonetheless revised the description for better understanding (as stated below). $L_{CA}$ is originally defined with the coverage indicator $k_i$ (0 or 1) but it cannot be used for gradient descent. So we use a soft version of $L_{CA}$, replacing $k_i$ by $\tilde{k}_i$ in equation (3.3). In the revised paper, we have made this point clear in Section 3 by adding equation (3.4) and the following explanations:
>
> “In order to run gradient-based methods, we replace the discrete indicator $(k_i, 1-k_i)$ in $L_{CA}$ with its soft version $(\tilde{k}_i, 1-\tilde{k}_i)$...(More in the paper).“
>
> **- 4. “There is a typo in the line immediately above Assumption 4.2 on p. 5.”**
>
> Thank you for pointing this out. We have revised accordingly.
>
> **- 5. “How difficult is it to tune the hyper-parameters? How sensitive is the method to the hyper-parameter choice?”**
>
> The hyper-parameters are tuned based on our tuning algorithm described in Appendix D, which is transparent and easy to implement. Our model does not seem sensitive to the hyper-parameters, as our training results are stable for hyper-parameters in a relatively large domain, and we can quickly locate the optimal hyper-parameter values via our tuning procedure.

---

> > ### Comment · AnonReviewer1 · 2020-11-25
> > **Follow-up**
> >
> > 2. This would be impossible for the real data sets, but could be done for the simulated data sets for which the conditional distributions are known. I agree that depending on the objectives of the paper, having the most relevant metric for evaluating statistical efficiency may be of lesser importance.
> > 3. $L_{CA}$ of Eq. (3.3) and $L_{CA}$ in the statement of Theorem 4.5 are off by a sign.

---

> ### Author Response · Authors · 2020-11-19
> **Response to Reviewer #1 (1/2)**
>
> We thank the reviewer for the valuable comments. We have tried to address the concerns of the reviewer, though we believe there may be some confusion from the reviewer regarding the problem setting and methodology in our work. We apologize for this confusion, and we have clarified as much as possible, as detailed below.
>
> **Weaknesses / Questions**
>
> **- 1.1 “It isn't clear to me at all how to set $\lambda_1$ to achieve a desired coverage level… What is unclear to me is whether the $\lambda_1=\lambda_1(a)$ that would achieve a fixed target marginal confidence level $1-\alpha$ is known or have to be estimated via some sort of a tuning procedure.”**
>
> $\lambda_1$ is a hyper-parameter in the loss function selected from the validation set on a random split (like the majority of hyper-parameters in deep learning algorithms). What perhaps is a bit different from standard tuning procedure is that we are validating against the empirical coverage probability, i.e., we select $\lambda_1$ such that the prediction level $1-\alpha$ is attained in the validation set.
>
> **- 1.2 “If $\lambda_1$ is obtained via tuning procedure, doesn't it make the method prone to overfitting?”**
>
> The tuning of $\lambda_1$, based on the validation set, is precisely used to avoid overfitting. This hyper-parameter, as well as others, are selected based on the training set and validated on the validation set in a tuning procedure. In this process we do not use the test set.
>
> Empirically, in our experiments, we do not find a huge performance gap between the training set and test set. Our model performs well on both the training dataset and the test set, and thus, we do not see a sign for overfitting. In addition, our approach is evaluated on multiple random splits and it is re-trained on each split with fixed hyper-parameters. The PI widths from those experiments are narrow and the standard deviations among different random splits are very small. It therefore appears that our model has robust performances on the datasets and does not run into overfitting issues.
>
> **- 1.3 “Also, doesn't it rather invalidate the experimental results, as the comparison methods use a pre-specified target level of 1-\alpha?”**
>
> We are not sure what exactly is the question from the reviewer here, but let us explain our results so that hopefully it would clarify the reviewer’s potential confusion. In Table 1, the target prediction level $1-\alpha=$ 95% is prescribed for all methods, and this information is used in the training process in our proposed approach and all the comparison approaches. Any trained model that has an empirical coverage on the test set at least 95% is regarded as providing a PI with a valid marginal coverage. Note that all the comparison algorithms also need to select the hyper-parameters via a similar tuning procedure, and therefore the comparisons are valid.
>
> **- 2. “Is it necessary to estimate the PI and the conditional coverage simultaneously?  Isn't it more interesting to estimate conditional coverage conditional on a particular output of L and U? therefore, estimate P after obtaining U and L?”**
>
> Indeed, we have thought of the “decoupled” approach suggested by the reviewer where one can first train a PI and then build on and adjust it for conditional coverage estimation. More precisely, this two-stage algorithm is implemented as: first, train the PI predictor with $L_{IW}$ and $L_{CP}$. Then, train another network to estimate the conditional coverage results using $L_{CA}$. This two-stage algorithm can be a possible choice for conditional coverage estimation if we already have a pre-trained PI predictor to begin with. However, the results are not as promising as ours. We have added the comparison results in Appendix E.3 to illustrate this point clearly.
>
> In addition, compared with this two-stage algorithm, our model has several advantages: our model is an end-to-end algorithm, which is very concise and easy to implement. In contrast, the two-stage algorithm involves two training procedures, with twice as many hyper-parameters needed to be selected for the two networks, and we have to train more neuron weights to get the PI predictor and the conditional coverage estimator.

---

> > ### Comment · AnonReviewer1 · 2020-11-25
> > **Follow-up questions**
> >
> > Thank you for the clarification. In hindsight, #1 was probably caused by a misreading of Algorithm 1. I take back most of #1. However, I believe it's still true that for the four comparison methods, the target coverage is more directly controlled, whereas in the proposed method, the value of $\lambda_1$ corresponding to the desired coverage level has to be found via a tuning procedure, such as Algorithm 1. If this is the case, then I think the difference is worth highlighting more (beyond what's already there in Section 5.3).
> >
> > Here are some additional questions:
> > 1. In theory, it looks like it ought to be possible to replace the predictor model with the QD-Ens of Pearce et al. (2018). This may allow the user a more direct control over the coverage. Would this approach be viable? Has it been tried? If so, what were the results? If not, why not?
> > 2. Thank you for the additional results of Appendix E.3. Could you provide the exact details of the hyper-parameter tuning procedure for the two-stage approach?
> > 3. If my understanding of the results is correct, the two-stage approach is not recommended for two reasons: (a) the resulting PIs undercover, and (b) ECE is substantially larger, mostly due to underestimation of the true conditional coverage probability. Do you have any explanation for the inferior performance?
> > 4. It would appear that it is a fairly straightforward problem to estimate the conditional coverage of a fixed (pre-estimated) PI based on a fresh batch of data (i.e., independent of the data that went into obtaining the given PI). The difficulty of the current problem appears to stem from having to both generate a PI and estimate its conditional coverage using the same set of data. Is this the case? If so, have you tried other data-splitting strategies? For example, what if one were to employ a three-way split, using the first partition to train the predictor module, the second to train the Ca-module, and the third to validate?
> >
> > To sum up, the simultaneous estimation of the PI and the conditional coverage continues to be the most puzzling aspect of the proposed method. The additional experimental results have only deepened the mystery.

---

### Official Review · AnonReviewer2 · 2020-10-28
**Review of Conditional Coverage Estimation for High-quality Prediction Intervals**

**Rating:** 7
**Confidence:** 4

**Review:**

##########################################################################

Summary:


In the paper, the author addresses a calibration-based conditional coverage error in order to avoid the difficulty of conditional coverage, which provides a middle ground between marginal (no conditional information) and conditional coverage (high computational cost). The author generates the idea building on prior work and designs a new loss function combining the high-quality criterion and a coverage assessment loss. The theoretical framework about the loss function is laid out clearly, and the performances on benchmark datasets provide accurate results which outperform the other baseline algorithms on high-quality prediction intervals generation.

Not directly estimating conditional coverage probability due to the challenge of approximating conditional distribution, the author develops a metric called calibration-based error to measure the estimation and its empirical counterpart is easy to compute. However, non-differentiable empirical counterpart of calibration-based error metric hinders the algorithm implementation when using gradient-based method, the author replaces it with Kullback-Leibler divergence and theoretically demonstrates this divergence is a tight upper bound of coverage error, which makes the optimization tractable.


##########################################################################

Reasons for score:


Overall, I vote for accepting. The theoretical study is carried out clearly and smoothly. The method proposed, which provides more opportunities for broad application, is creative and well-demonstrated. My major concern is about the clarity of the paper in terms of the order of the presentation, and the possibility of a decoupled method. Hopefully the authors can address my concern in the rebuttal period.


##########################################################################

Pros:


1. This paper considers calibration of the coverage incidence, which is important information for real deployment of any prediction interval method.

2. The proposed loss is clear, intuitive and easy to understand.

3. Nice concentration bound for the calibration part.



##########################################################################

Cons:


1. Lemma A.2 (a) is an important argument for constructing the subsequent error metric. I think the statement appears to be too strong in general therefore can not be achieved in practice.

2. The authors should make it crystal clear that they are not proposing a PI with conditional coverage guarantee; instead, just try to provide an estimate of the conditional coverage guarantee given the feature. Somehow I had the impression that the goal was the former, until late in the paper.

3. The methodology in Section 3 is not naturally motivated from the many discussions and definitions about the coverage estimation error starting from Definition 2.1 to the end of Section 2. Their connection is really in the back-end which is not shown until Section 4. It may be better to reshuffle the order of the presentation, or adding some explanation in Section 3 when Loss_CA is introduced. The main gap is that one cannot see how Loss_CA should be defined as in (3.3) after reading all these discussion about CE in Section 2.

4. Is there any advantage to consider the PI problem and the calibration problem in the same network? What is wrong with first estimate the PI, and then estimate the coverage incident given the PI as a separate problem? In this de-coupled framework, Loss_IW and Loss_CO will be used in the PI problem and Loss_CA will be used in the calibration problem, and they do not need to share the same network. More to the point: when a shared total loss is the goal of minimization, it would seem that the PI would evolve to make the task of calibration easier (to have small calibration error); there may be some cost to pay for this joint training, either in terms of IW or coverage. However, it is not clear to me what price we have to pay in the joint problem in (3.4)

 5. Table 1: why 1.13 in the second row is bold face when the IW for QD_Ens has a smaller IW of 1.09?

##########################################################################

Questions during rebuttal period:

Please address my comments in cons above plus the followings.

For the total loss (3.4), since the non-coverage probability is incorporated into the loss, we have to manually or adjust these tune parameters until the marginal coverage probability is satisfied. This does not seem to be trivial. Will this reduce the computation efficiency? Will a constrained optimization instead be faster than this? What is the hyper-parameter $\lambda_3$ used for?

The proof on VC-dimension and coverage assessment approximating Kullback-Leibler divergence are well-organized and detailed. However, perhaps supplying the proof on excess risk would be helpful too.


####

Minor comments:

I think a useful insight that the author can consider is that the calibration problem can be viewed as a classification problem in which the response is the event that the PI covers the Y value (\ind{ Y \in PI}). Then it would be class that the total loss is a PI problem joint with a classification problem.

---

> ### Author Response · Authors · 2020-11-19
> **Response to Reviewer #2 (2/2): Questions during Rebuttal Period**
>
> **Questions during Rebuttal Period**:
>
> **- “For the total loss (3.4), since the non-coverage probability is incorporated into the loss, we have to manually or adjust these tune parameters until the marginal coverage probability is satisfied. This does not seem to be trivial. Will this reduce the computation efficiency? Will a constrained optimization instead be faster than this? What is the hyper-parameter $\lambda_3$ used for?”**
>
> Thanks to our tuning procedure in Appendix D, we can easily pick up these hyper-parameters. Thus, it will not reduce the computation efficiency. Our loss function is in disguise motivated by a constrained optimization problem:
>
> Minimize $L_{IW}$, subject to $L_{CP}\ge c_1$ and $L_{CA}\le c_2$
>
> In the formulation above, ECE is replaced by $L_{CA}$ as the former cannot be readily used for gradient descent. To handle this constrained optimization problem, we utilize a Lagrangian formulation that leads to our total loss function, on which we minimize over the neural network. $\lambda_3$ is inside the sigmoid loss for $L_{CP}$ (equation 3.2), where the sigmoid loss is introduced as a smooth version of the 0-1 loss.
>
> **- "The proof on VC-dimension and coverage assessment approximating Kullback-Leibler divergence are well-organized and detailed. However, perhaps supplying the proof on excess risk would be helpful too."**
>
> We thank the reviewer for the compliment on the organization and details of our proof! However, we would like to clarify from the reviewer on the meaning of “excess risk”, and we are happy to revise or add on our proof according to the clarified suggestion.
>
> **Minor Comments**:
>
> We appreciate this comment, and completely agree with this insight on the connection with classification. In fact, our approach is motivated from the calibration concept in classification precisely based on this insight (as described in the second paragraph in Introduction).

---

> ### Author Response · Authors · 2020-11-19
> **Response to Reviewer #2 (1/2)**
>
> We thank the reviewer for the many helpful comments, and we are very excited to hear the positive opinion of the reviewer regarding our contributions. We have revised the paper to improve readability and clarify claims to address the reviewer’s comments, as detailed below.
>
> **Cons**
>
> **- “Lemma A.2 (a) is too strong in general therefore can not be achieved in practice.”**
>
> We agree with the reviewer. Indeed, this statement is strong and is motivated from a theoretical point of view. Nonetheless, it gives a guidance that in order to well resemble the conditional coverage, an estimator should be perfect-calibrated on as many subsets on the feature space as possible. We have now added explanations in the appendix Section A.3 in the paper:
>
> “Lemma A.2 (a) is motivated from a theoretical point of view. It provides a guidance that in order to well resemble the conditional coverage, an estimator should be perfect-calibrated on as many subsets on the feature space as possible.”
>
> **- “The authors should make it crystal clear that they are not proposing a PI with conditional coverage guarantee; instead, just try to provide an estimate of the conditional coverage guarantee given the feature. Somehow I had the impression that the goal was the former, until late in the paper.”**
>
> We apologize for this confusion. We have added our goal explicitly in Section 1 (end of the first paragraph):
> “Our main goal is to meaningfully incorporate and assess conditional coverages in high-quality PIs.”
> where we use “incorporate” and “assess” to refer later to the addition of the $L_{CA}$ loss in training and the ECE metric respectively.
>
> **- “The methodology in Section 3 is not naturally motivated from the many discussions. Their connection is really in the back-end which is not shown until Section 4. It may be better to reshuffle the order of the presentation, or adding some explanation in Section 3 when Loss_CA is introduced. ”**
>
> We thank the reviewer for this great suggestion. We have now added explanations on the implications of $L_{CA}$ loss function in Section 3 with:
>
> \>1. Additional explanations on the $L_{CA}$ loss before equation (3.3):
> “Associated with the Ca-Module, we introduce a coverage assessment loss $L_{CA}$ to estimate the conditional coverage.”, and
>
> \>2. Reference to the developments in Section 4 after equation (3.3):
> “We will show in Section 4 that the expectation of coverage assessment loss $L_{CA}$ provides an upper bound for both conditional coverage error (Definition 2.1) and calibration-based conditional coverage error (Definition 2.3). Hence, minimizing $L_{CA}$ contributes to the recovery of the conditional coverage.”
>
> **-  “Is there any advantage to consider the PI problem and the calibration problem in the same network?”**
>
> We have indeed considered the same idea as the reviewer. We implemented this decoupled framework in two stages: first, train the PI predictor with $L_{IW}$ and $L_{CP}$. Then, train another network to estimate the conditional coverage results using $L_{CA}$. The results from this two-stage algorithm are not always as promising as our proposed model. We have added the comparison results with this two-stage method in Appendix E.3 to make this point clear.
>
> In addition, we would like to highlight that, compared with this two-stage algorithm, our approach has several advantages: Our model is an end-to-end algorithm, which is very concise and easy to implement. On the other hand, the two-stage algorithm involves two training procedures, with twice as many hyper-parameters needed to be selected for the two networks, and extra work to figure out the architecture for the second network. We also have to train more neuron weights for the PI predictor and the conditional coverage estimator. Overcoming these drawbacks is one motivation for our integrated architecture, and we empirically demonstrate that our approach achieves superior performances on both PI generation and conditional coverage estimation. However, we should note that, if we are already given a pre-trained PI predictor to begin with, the decoupled algorithm could be a possible choice for conditional coverage estimation. We have added the two-stage algorithm, the discussions above, and comparison results (as mentioned before) in Appendix E.3.
>
> **- “Table 1: why 1.13 in the second row is bold face when the IW for QD_Ens has a smaller IW of 1.09?”**
>
> We have employed the criteria in Pearce et al. to evaluate the performance of PIs: the best IW is achieved by the model with the smallest IW value among those with $CP \ge$ 95% (Section 5.2). In Table 1, the CP value for QD_Ens is 0.94, which is less than the target prediction level. For better understanding, we have added the following in Table 1’s caption:
> “The best IW results, marked in bold, are achieved by models with the smallest IW value among those that meet the 95% prediction level.”

---

### Official Review · AnonReviewer3 · 2020-10-28
**Useful contributions but disconnected, main ideas not explored theoretically or experimentally**

**Rating:** 4
**Confidence:** 4

**Review:**

After author response:

I disagree with the discussion on MSE. For the empirical estimator you mention, we have:
$$E[(Y - \hat{P}(X))^2] = E[(Y - A(X))^2] + E[(A(X) - \hat{P}(X))^2]$$
Importantly, $E[(Y - A(X))^2]$ is a fixed value regardless of what $\hat{P}$ you use. So while you can’t compute $E[(A(X) - \hat{P}(X))^2]$, you can compare whether this is higher or lower for a particular $\hat{P}$ by just comparing $E[(Y - \hat{P}(X))^2]$.

By the way, this is directly analogous to classification. In classification, Y | X is stochastic, it is 1 with some probability A(X) and 0 with probability 1 - A(X). Indeed, we cannot measure $E[(A(X) - \hat{P}(X))^2]$ directly - instead we estimate $E[(Y - \hat{P}(X))^2]$, but that’s just off by some fixed value (which does not depend on $\hat{P}$).

At a higher level, there isn’t really a distinction between classification and the setting here. Let f(X) be your confidence interval, and introduce a random variable A given by A = 1 if Y \in f(X) and A = 0 if Y \not\in f(X) be a random variable, then we are precisely estimating P(A = 1 | X). This exactly corresponds to classification, where the label A is either 0 or 1, and we are estimating P(A = 1 | X).

As such, it’s important to compare with standard baselines (e.g. the 2 stage approach). Use the neural network features instead of training the coverage estimation model from scratch in the second stage, and show the MSE and calibration error values.

I still think it’s unclear there is much interaction between the “high quality” confidence interval and coverage estimation. As the author response says, setting $\lambda_2 = 0$ and turning off the Ca-module, would not affect the confidence intervals produced.

#########################################################################

Summary:

This paper tackles two problems:
1. Providing high quality prediction intervals for regression problems. In particular, they want prediction intervals that have a desired marginal coverage (e.g. true output is in prediction interval 95% of the time), and average interval width is small.
2. Estimating the coverage of a prediction interval (conditional coverage estimation).

For (2) they propose measuring the calibration of the coverage estimator. They propose training (1) and (2) jointly using a sum of 3 losses. On the theoretical side, (a) they show that the log loss upper bounds the calibration error motivating its use as a surrogate loss, and (b) they show that given enough data objectives (1) and (2) can be trained jointly with low generalization error on the calibration error. They show experimentally that their approach mostly gets smaller interval widths for the same coverage level, than prior work.

#########################################################################

Reasons for score:

The paper has a lot of interesting ideas, but they seem rather disconnected to me. A key missing ingredient is that the paper does not explain why jointly estimating the coverage improves the quality of prediction intervals. Their theory only motivates that they can estimate the conditional coverage. On the experimental side, they don’t have ablations without the coverage estimation (that is, with only losses L_IW and L_CP in their notation) to check whether the coverage estimation loss L_CA helps. I’m unconvinced about the experimental protocol (more details below), the setup and architectures seem different from Pearce, and it’s unclear if results are from a single split which hyperparameters are tuned on. On the plus side, the method seems to have narrower intervals so could be useful for practitioners if some of these concerns are ironed out.

I believe this work could have solid contributions if these issues are cleared up and the paper is made cohesive, and my assessment is based on the current state of the paper as opposed to the research direction. Keep up the good work!

#########################################################################

Pros:

- The idea of outputting not only an interval but also a coverage estimate sounds interesting and potentially useful, e.g. it can allow us to identify cases where the intervals do not have the desired coverage. Measuring calibration of the coverage estimator makes sense (it is weaker than a pointwise guarantee, but stronger than a marginal guarantee).

- The (L_IW and L_CP) loss used to train prediction intervals seems sensible. It looks related to Rosenfeld et al, but uses a sigmoid instead of a hinge to penalize predictions that fall out of the prediction interval. Intuitively this makes sense to me for neural nets, since anecdotally my experience is that sigmoid style losses work better. Although if using softmax instead of hinge is being positioned as a major point (I didn’t think it was) there should be a comparison with hinge loss.

- This paper seems to get better results than prior work, which is definitely a positive.

- I skimmed the proof of Theorem 1 and it looks correct, and Theorem 2 sounds likely true.

#########################################################################

Cons:

- The main missing ingredient is the connection between predicting coverage and getting tighter intervals. Why does predicting the coverage (the L_CA loss) make the intervals tighter? Taking a step back, does it even make the intervals better? The theory does not address this, and there aren’t any experiments that this L_CA component specifically helps. My judgement (not in the paper) is that the L_CA loss is indeed lower if the prediction intervals have high coverage (if \hat{k}_i is close to 1 and \hat{P} is accurate). However, the L_CP term already encourages high coverage, so it’s unclear if L_CA is doing anything. If I only use L_IW and L_CP can I get the same results? Would need to do grid search to choose the right hyperparameters lambda_1 and lambda_3, which would be different after removing L_CA.

- Experimental protocol is unclear. Peirce et al do 20 random splits into 90% train - 10% test, and report means and standard deviations. This paper says the split is 80% train - 20% test. Is there just one split, otherwise what are the standard deviations? The numbers for Peirce et al are quoted directly from their paper as well, so under a different setting. It looks like multiple hyperparameters (lambda1, lambda2, lambda3) are being tuned on the same validation set that the final results are measured on?

- Experiments use different models from prior work and make multiple changes to the losses (sigmoind instead of hinge) and architectures. For example, Peirce et al use one hidden layer and 50 nodes (except for Protein and Song Layer where they seem to use 100). Where are the gains really coming from? Is it just from making the network deeper?

#########################################################################

Questions and things to improve:

I’ll certainly reconsider my score if at least some of the above are addressed, especially 1. the experimental setup needs to be clarified (hyperparameter tuning, use 20 random splits, report std-devs, 80%-20% split), 2. Need to have ablations without coverage prediction (tune lambda1 and lambda3 in this case), to see if it actually helps. The paper oversells a little in claims of “theoretically justified” since it only justified why the coverage assessment is accurate, but not why coverage assessment helps get better intervals - this needs to be edited.

#########################################################################

Additional comments on theory:

- Theorem 1: shows log loss is an upper bound for cal error. It shows that the log loss to a suitable power upper bounds the lp calibration error. This looks like a nice and useful result, and I at least haven’t seen this before. I skimmed the proof and it looks correct. The naive way to upper bound say the l2 calibration error is using the MSE, but it does look like the log loss is tighter in many interesting cases. I could see this being positioned as a more important contribution of the paper if there is some argument (or examples motivating) for why the bound is tighter than alternatives (like MSE for l2 calibration error)

- Theorem 2: VC dim argument to show that log loss is approximated correctly by finite samples. I haven’t studied the proof of this theorem. My main questions are 1. An alternative to joint training is you can train the lower bound and upper bound estimator first, and then train the coverage module later on held out data. That should also have an exponential rate, with C* being a function of V_0. The theorem doesn’t seem to explain the advantage of joint training vs the split procedure?, 2. If we just want an exponential tail bound, we could also use a parameter space eps-cover, e.g. assuming the neural network is say L-Lipschitz. What’s the advantage of the VC dimension argument?

---

> ### Author Response · Authors · 2020-11-19
> **Response to Reviewer #3 (5/5): Additional Comments on Theory**
>
> **Additional Comments on Theory**:
>
> **1. “Theorem 1: shows log loss is an upper bound for cal error …  for why the bound is tighter than alternatives (like MSE for l2 calibration error)”**
>
> We thank the reviewer for the positive comment regarding this result. Indeed, this result is new and we have not seen any similar results before.
>
> Regarding the reviewer’s thought about using MSE, if we understand correctly, this is precisely the conditional coverage error metric $\widetilde{CE_2}$ (with $L^2$ norm) in the theorem. Note, however, that this metric cannot be used easily for training, which is the reason why we propose the Kullback-Leibler-type loss. In contrast to $L_{CA}$ which is unbiased and provides guaranteed estimation accuracy for the Kullback-Leibler-type loss as we prove in Theorem 4.5, an unbiased estimation for MSE, i.e., $\widetilde{CE_2}$ , requires refined information on the conditional coverage itself. We have given more explanations about this and an example in the paper at the end of Appendix A.4:
>
> “Finally, we give some explanations about why the conditional coverage error $\widetilde{CE_p}$ cannot be used easily for training. Unlike $L_{CA}$ which is unbiased and provides guaranteed estimation accuracy for $\mathbb{E}[K_1(X)]$ (Theorem 4.5), it is in general not easy to establish an empirical calculation for $\widetilde{CE_p}$. Take $\widetilde{CE_2}$ as an instance. A heuristic argument is to use...(More in the paper).”
>
> Also, as the reviewer suggested, we now emphasize more our theoretical contributions in Theorem 4.1, by adding underneath Theorem 4.1:
> “The type of results in Theorem 4.1 that bounds an $L^p$ conditional coverage error via a Kullback-Leibler-type error is new as far as we know.”
>
> We have also added in the last paragraph of Introduction:
> “by developing concentration bounds relating the coverage assessment loss and conditional coverage error”.
>
> **2.1 “The theorem doesn’t seem to explain the advantage of joint training vs the split procedure?”**
>
> Theorem 4.5 intends to theoretically explain the effectiveness of the CA-loss for conditional coverage estimation. Split procedure is also a possible way for conditional coverage estimation if we already have a PI predictor to begin with. However, if we construct a PI from scratch, then this latter approach’s performance is not always as promising as our algorithm. We have added the comparison results in Appendix E.3 to demonstrate this point.
>
> In addition, compared with this two-stage algorithm, our approach has several advantages: ours is an end-to-end algorithm, which is very concise and easy to implement. The two-stage algorithm involves two training procedures, with twice as many hyper-parameters needed to be selected for the two networks and moreover requires training more neuron weights to get the PI predictor and the conditional coverage estimator.
>
> **2.2 “What’s the advantage of the VC dimension argument?”**
>
> We agree that L-Lipschitz is an alternative approach to obtain concentration bound for neural networks. However, it may require additional assumptions on network parameters, such as assuming a uniform bound on all network parameters. In contrast, using VC dimension argument does not require assumptions on network parameters (Bartlett et al., 2019).

---

> ### Author Response · Authors · 2020-11-19
> **Response to Reviewer #3 (4/5): Clarifications on Experimental Setup (2/2)**
>
> **Clarifications on Experimental Setup (2/2)**:
>
> **- Standard deviation results and random splits**
>
> The reviewer rightly asked about the standard deviation and random splits. We have indeed followed the approach in Pearce et al., and we apologize for the unclearness in the writing. For the high quality criterion, our $L_{CP}$ is the same as Pearce et al., which also uses the sigmoid function. The main difference is the ensemble method, in which we remove the standard deviation terms as explained above. We only run 5 random splits with fixed hyper-parameters and for conciseness, we did not report the standard deviations in the paper. However, these are readily obtainable, and the standard deviations for 5 random splits with fixed hyper-parameters are as follows:
>
> **Mean and standard deviations from 5 random splits with fixed hyper-parameters**
>
>
> | Datasets      |      CP  |  IW   |
> |:------------------|:-------------:|:-----------------:|
> | Boston          |0.95+/-0.01      |1.04+/-0.02     |
> | Concrete      |0.95+/-0.00       |1.13+/-0.03     |
> | Kin8nm         |0.95+/-0.00      |1.04+/-0.02     |
> | Plant             |0.95+/-0.00       |0.84+/-0.01     |
> | Protein         |0.95+/-0.00       |2.26+/-0.03    |
> | Wine             |0.95+/-0.01       |2.59+/-0.04  |
> | Yacht            |0.98+/-0.02       |0.16+/-0.01|
>
> As suggested by the reviewer, we have also conducted more experiments on 20 random splits with fixed hyper-parameters (same procedure as the 5 random splits but increase the number of random splits) to further demonstrate the robustness of our approach. Due to time limit, we have only finished a couple of the datasets and their results are as follows:
>
> **Mean and standard deviations from 20 random splits with fixed hyper-parameters**
>
> | Datasets      |      CP  |  IW   |
> |:------------------|:-------------:|:-----------------:|
> | Boston         | 0.95+/-0.01    |  1.03+/-0.02 |
> | Concrete      |  0.95+/-0.01  |    1.14+/-0.03  |
>
> The results from 20 random splits are consistent with our results obtained from 5 random splits, showing the robustness of our PI predictor. We will incorporate these results (along with the rest of the datasets) into the paper if the reviewer deems necessary to do so.
>
> **-  Different model architectures**
>
> We use different model architectures and hyper-parameters from Pearce et al. The main reason is that we have incorporated an additional $L_{CA}$ loss in our training, and also we have our new ensemble strategy. To account for these, we optimize our architecture that is different from Pearce et al. Note that it would not be fair to apply our new architecture to Pearce et al.’s setting, because (we believe) they have optimized their architecture for their purpose and using our architecture on them would likely penalize their performances. In addition, as explained above, to meaningfully evaluate our ECE results, we use a larger share of test set size in the split. Given all these considerations, to give a comparison as fair as possible we directly quote Pearce et al.’s results.

---

> ### Author Response · Authors · 2020-11-19
> **Response to Reviewer #3 (3/5): Clarifications on Experimental Setup (1/2)**
>
> **Clarifications on Experimental Setup (1/2)**:
>
> **-  80% train - 20% test split**
>
> Our choice of 80%-20% split, which differs from Pearce et al.’s 90%-10% split, is motivated from the need to increase the test size in order to get a meaningful ECE evaluation. As specified in Section 2 (Equation (2.4)) and Appendix A.4 (Equation (A.6)), ECE is evaluated based on dividing [0,1] into M sub-intervals. The larger M is, the more precise is in using ECE to approximate CE, the latter being the ideal conditional coverage error estimator. On the other hand, a larger test set size is needed to support the use of a larger M without deteriorating the statistical quality of the ECE. This delicate tradeoff motivates us to increase the share of the test set in our split.
>
> Moreover, even though our model is trained under less training data than Pearce et al., it still leads to comparable PI prediction results with baseline algorithms, which is a favorable conclusion to observe. We have now added the following discussion into Section E.1 to clarify our choice of split:
>
> “The choice of 80%/20% split, compared to the 90%/10% split in Pearce et al., is motivated from the need to increase the test size in order to get a meaningful ECE evaluation. The latter is due to that evaluating ECE requires binning, where using a larger number of bins approximates more closely CE, but also requires a larger test size to sustain enough statistical quality for the resulting ECE estimate. This delicate tradeoff motivates us to increase the share of the test set in our split.”
>
> **- Ablations without L_CA and hyperparameter tuning**
>
> We have 3 tunable parameters $\lambda_i$, $i= 1,2,3$ in our loss function with $\lambda_1$ and $\lambda_3$ for high quality PI generation ($L_{IW}$ and $L_{CP}$), and $\lambda_2$ for conditional coverage estimation ($L_{CA}$ loss). We assign a very small value (~10^-5) for $\lambda_2$ so that the $L_{CA}$ loss will mainly function at the CA-Module for conditional coverage estimation. Therefore, removing $L_{CA}$ loss will not have a huge impact on the high quality criterion. However, $L_{CA}$ loss is the only loss for the Ca-Module, which is critical for conditional coverage estimation - That is, without this loss we cannot obtain any conditional coverage estimation (which is our main goal in this paper that is unaddressed by previous works), and hence we cannot remove it.
>
> Following previous research (Pearce et al., 2018), our hyperparameters ($\lambda_1, \lambda_2, \lambda_3$) are selected using the validation set from a random split. Then, they are fixed during the training and evaluation on other random splits. In Appendix D, we suggest an easy-to-implement tuning procedure that could quickly guide the search of optimal hyperparameters.
>
> We have now added the following details in Appendix E.1:
>
> “Following Pearce et al. (2018), our hyper-parameters are selected using the validation set from a random split. Then, they are fixed during the training and evaluation on other random splits.”

---

> ### Author Response · Authors · 2020-11-19
> **Response to Reviewer #3 (2/5): Better Prediction Interval Results**
>
> **Better Prediction Interval Results**:
>
> The reviewer is rightly puzzled why we can generate tighter intervals than previous works. Indeed, our main purpose in this paper, as stated above, is to add conditional coverage information into the PI, and has no direct relation to improving the general tightness of the intervals. However, we believe the ensemble method used during our implementation contributes to the tighter intervals. In Pearce et al., the authors enlarge the intervals by adding/subtracting the 1.96*standard deviation on the upper/lower bound obtained from the averages (their equation 21), i.e.,
>
> $\bar{y}_{U_i}=\frac{1}{m} \sum_{j=1}^{m} \hat{y}$ _ {$U_{ij}$}, (Eq 19)
>
> $\hat{\sigma}^2_{model} = \sigma^2_{U_i}= \frac{1}{m-1}\sum_{j=1}^{m} (\hat{y}$ _ {$U_{ij}$} $-\bar{y}_{U_i})^2$, (Eq 20)
>
> $\tilde{y}_{U_i} =\bar{y}_{U_i} + 1.96\sigma_{U_i} $, (Eq 21)
>
> “A similar procedure is followed for $\tilde{y}_{L_i}$, subtracting rather than adding $1.96\sigma_{L_i}$.” (Pearce et al, 2018)
>
> Our implementation does not add these adjustment terms, i.e., we use
>
> $Lower\ bound\qquad \qquad  \bar{L}:=\sum_{i=1}^{m} \frac{1}{m}L_i,$
>
> $Upper\ bound\qquad \qquad  \bar{U}:=\sum_{i=1}^{m} \frac{1}{m}U_i,  \qquad \qquad         $  (Eq 3.6)
>
> $Coverage\ estimator\ \ \ \bar{\hat{P}}:=\sum_{i=1}^{m} \frac{1}{m}\hat{P}_i.$
>
> To explain our rationale, note that while the standard deviation adjustments may look reasonable at first glance, the choice of $\lambda$’s in the loss function against the empirical coverage should take care of most uncertainties regarding the fluctuation of the coverage estimates, and thus we do not see a strong evidence to support the “mean +/- 1.96*standard deviation” adjustments. Indeed, empirically we did not see such an adjustment could improve results. Moreover, the procedure of using simply the mean for deep ensemble has been applied by other well-cited papers such as Lakshminarayanan et al. (2017). Given our reasoning, our empirical findings, and these other papers, we thus choose to use simple means in our proposed implementation, which turns out to perform better than Pearce et al.

---

> ### Author Response · Authors · 2020-11-19
> **Response to Reviewer #3 (1/5): Clarifications on the Purpose of Introducing $L_{CA}$ Loss**
>
> We greatly thank the reviewer for the many helpful suggestions. We also appreciate the positive feedback on our contributions, which is encouraging. We have revised the paper to enhance clarities on our problem of interest and methodology. In the following, we address the reviewer’s concerns in several main categories:
>
> **Clarifications on the Purpose of Introducing $L_{CA}$ Loss**:
>
> A main concern of the reviewer is the apparent disconnection between “local” coverage estimation and tighter interval. To address this, let us first clarify our contribution, which is to propose an end-to-end framework to provide accurate conditional coverage estimation that is critical for downstream decision-making but unassessed in previous works. In other words, while tight intervals are certainly desirable, if they are constructed based only on marginal coverage (as concretized in the constrained optimization in equation (2.1)), they may be way off in terms of conditional coverage given a specific feature value. This latter issue is the main question to address in this paper.
>
> To achieve our goal above, we design the $L_{CA}$ loss specifically for conditional coverage estimation. If it is removed, the algorithm will construct a PI only according to the criterion in equation (2.1), but with no accountancy or information on its conditional coverages.
>
> With this in mind, our theoretical results are developed to show how minimizing $L_{CA}$ is related to minimizing the conditional coverage error.
>
> We apologize for the confusions to the reviewer regarding the points above. We have now added the following sentences in both the introduction (Section 1) and the methodology (Section 3) to clarify our problem setting and its relation to the $L_{CA}$ loss:
>
> \- We have stated our goal explicitly in Section 1 (end of the first paragraph):
> “Our main goal is to meaningfully incorporate and assess conditional coverages in high-quality PIs.”
> where we use “incorporate” and “assess” to refer later to the addition of the L_CA loss in training and the ECE metric respectively.
>
> \- We have added explanations on the $L_{CA}$ loss in Section 3, before equation (3.3):
> “Associated with the Ca-Module, we introduce a coverage assessment loss $L_{CA}$ to estimate the conditional coverage.”
>
> and after equation (3.3):
> “We will show in Section 4 that the expectation of coverage assessment loss $L_{CA}$ provides an upper bound for both conditional coverage error (Definition 2.1) and calibration-based conditional coverage error (Definition 2.3). Hence, minimizing $L_{CA}$ contributes to the recovery of the conditional coverage.”
>
> We hope these changes would make our goal, motivation for adding the $L_{CA}$ loss, and theoretical contributions clear.

---

> > ### Comment · AnonReviewer3 · 2020-11-23
> > **Thanks for clarifying main goal is producing coverage estimates, and getting better intervals is tangential**
> >
> > Thanks for clarifying and editing the paper - in the original version it seemed like both getting good intervals and estimating their coverage were equally important goals. It seems now that the main goal is estimating coverage. I think this is a somewhat less interesting goal since it's a more standard problem, but a thorough investigation of competing methods for this could be useful for the ICLR community.
> >
> > A few technical points:
> > 1. If the main goal is to estimate coverage, then that's what the experiments should focus on. In the main paper, there is no comparison with alternative ways to estimate coverage. Table 2 only shows the ECE for the proposed method but no baselines. Appendix E.3 has a comparison on one dataset, but there are minimal details on the baseline approach, what hyperparameters were tuned, etc. Additionally, in the two stage method you don't need to train an entire new network from scratch to estimate coverage - you can just train a linear layer on top of the penultimate layer. Instead, the central experiment seems to be comparing the interval quality of other methods (Table 1), which from the author response seems like isn't the goal of the paper.
> > 2. Calibration in itself is quite a weak metric for coverage assessment. For example, if we always predict the marginal coverage probability, then we will be perfectly calibrated. In the calibration literature they typically also show the MSE of predictions to indicate the sharpness.
> > 3. As for why their method gets better interval quality---this seems unrelated to the CA-module, but because they use an assortment of other techniques (ensembling, computing the intervals differently, etc). So their improvements on table 2 seem to be coming for different reasons. These need to be separated better. That's why I suggested getting the numbers in Table 1 when lambda is set to 0 (without CA-module). You can still get the numbers in Table 1 (CP and IW).
> >
> > Some parts of the writing and the title should also be clarified. Conditional Coverage Estimation for High-quality Prediction Intervals seems like "Conditional Coverage Estimation" is an approach to achieve "High Quality Prediction intervals", when it seems like this is not the case. Does "high-quality" matter for the problem being studied here? Shouldn't the proposed method work for any prediction interval method - or do baselines specifically work poorly at estimating coverage when the prediction intervals are high quality? Why can't we just say "Conditional Coverage Estimation for Prediction Intervals" and remove the mention of "high quality" everywhere in the paper?

---

> > > ### Author Response · Authors · 2020-11-25
> > > **Response to Reviewer #3 (3/3)**
> > >
> > > **- 3 “As for why their method gets better interval quality---this seems unrelated to the CA-module, but because they use an assortment of other techniques (ensembling, computing the intervals differently, etc). So their improvements on table 2 seem to be coming for different reasons. These need to be separated better. That's why I suggested getting the numbers in Table 1 when lambda is set to 0 (without CA-module). You can still get the numbers in Table 1 (CP and IW).”**
> > >
> > > If we set the $\lambda_2 = 0$ (remove the Ca-module) in Table 1, the results appear to be minimally affected in terms of CP and IW. This is because we have assigned a very small value (~10^-5) for $\lambda_2$ so that the $L_{CA}$ loss will mainly function at the Ca-Module for conditional coverage estimation. We are happy to properly document and show these in the paper, but are constrained by the time limit in this rebuttal stage...
> > >
> > > **- “Some parts of the writing and the title should also be clarified. Conditional Coverage Estimation for High-quality Prediction Intervals seems like "Conditional Coverage Estimation" is an approach to achieve "High Quality Prediction intervals", when it seems like this is not the case. Does "high-quality" matter for the problem being studied here? Shouldn't the proposed method work for any prediction interval method - or do baselines specifically work poorly at estimating coverage when the prediction intervals are high quality? Why can't we just say "Conditional Coverage Estimation for Prediction Intervals" and remove the mention of "high quality" everywhere in the paper?”**
> > >
> > > Indeed, our Ca-Module can provide conditional coverage estimation for any prediction interval. However, this wouldn’t be as meaningful because without the high-quality criterion (i.e., the width minimization objective) one has too much freedom in the interval width to achieve any required (conditional) coverage level.
> > >
> > > As we emphasize at the beginning of this response, the major challenge and the motivational situation is when one needs to balance the high-quality criterion with the conditional coverage estimation, which plays simultaneous roles in the loss function in the training. In other words, our major goal is indeed to address challenges in constructing PIs that perform well both in overall width and in conditional coverage.

---

> > > ### Author Response · Authors · 2020-11-25
> > > **Response to Reviewer #3 (2/3)**
> > >
> > > **- 2.1 “In the calibration literature they typically also show the MSE of predictions to indicate the sharpness.”**
> > >
> > > We thank the reviewer for bringing up the use of MSE from classification calibration. Indeed, we have thought about this carefully in our investigation, and we realized there is a subtle but important issue when using this notion in estimating conditional coverage in regression. That is, the use of MSE is not as straightforward as what the reviewer may think.
> > >
> > > First, in classification (e.g., Kumar et al., 2019; Kuleshov & Liang, 2015), the definition of MSE (e.g., Definition 2.2 in Kumar et al., 2019) is $MSE:= E[(\hat{P}(X) - Y)^2]$, where $Y\in$ {0,1} is the ground-truth label in binary classification. In our conditional coverage problem for regression, the natural definition of MSE is $MSE:= E[(\hat{P}(X) - A(X))^2]$, where $A(X)$ is the ground-truth label for conditional coverage. This is precisely $\widetilde{CE}^2_2$ in our Definition 2.1. Note that here $A(X)$ is a real-valued unknown.
> > >
> > > A possible empirical estimator for $\widetilde{CE}^2_2$, which mimics the one used for MSE in classification, is
> > >
> > > $\hat{\theta}:=\frac{1}{n}\sum_{i=1}^{n} (k_i-\hat{P}(x_i))^2$
> > >
> > > where $k_i= 0 \text{ or } 1$ indicates whether each data point has been captured by the PIs. (This is the same estimator to evaluate MSE in the classification literature.)
> > > Unfortunately, $\hat{\theta}$ is a systematically biased estimator for $\widetilde{CE}^2_2$, i.e., this bias does not vanish as the sample size $n$ grows. This can be seen by
> > >
> > > $\mathbb{E}[\hat{\theta}]=\mathbb{E}[|A(X)-\hat{P}(X)|^2]+\mathbb{E}[A(X)-A(X)^2]$
> > >
> > > which is the target MSE(i.e., $\widetilde{CE}^2_2$) plus an extra bias term that is difficult to estimate in general.
> > >
> > > Thus, $\hat{\theta}$ is not a well-designed evaluation metric for the conditional coverage estimator. A large $\hat{\theta}$ value does not necessarily indicate weak performance of the conditional coverage estimator $\hat{P}$ since this might come from the second bias term. For example, suppose the ground-truth conditional coverage $A(x)\equiv 0.5$. Then $\hat{\theta}=0.25$ even if $\hat{P}(x)=A(x)$ is exactly the ground-truth conditional coverage.
> > >
> > > We refer the reviewer to further explanations in Appendix A.4 (also mentioned in our previous rebuttal):
> > >
> > > “Finally, we give some explanations about why the conditional coverage error $\widetilde{CE}$ cannot be used easily for training. Unlike $L_{CA}$ which is unbiased and provides guaranteed estimation accuracy for $\mathbb{E}[K_1(X)]$ (Theorem 4.5), it is in general not easy to establish an empirical calculation for $\widetilde{CE}_p$. Take $\widetilde{CE}_2$ as an instance. A heuristic argument is to use...(More in the paper).”
> > >
> > >
> > > **- 2.2 “Calibration in itself is quite a weak metric for coverage assessment.”**
> > >
> > > Given the challenge in using the standard MSE presented above, we propose calibration-based conditional coverage error since it provides a middle ground between the enforcement of marginal coverage (lacking any conditional information) and conditional coverage (computationally intractable). We understand there may be better metrics for conditional coverage assessment, and also estimates for MSE kindly suggested by the reviewer, in the future (perhaps built on this work), but as of now, there is no alternative to assess conditional coverage in regression as we are aware of.
> > >
> > > In this paper, we also strive to provide more insights in using calibration as a metric. We have shown in Lemma A.2 (a) that a coverage estimator is the conditional coverage if and only if it is a perfect-calibrated coverage estimator on any positive-probability measurable subset of $\mathcal{X}$. This result provides a guidance that in order to well resemble the conditional coverage, an estimator should be perfect-calibrated on as many subsets on the feature space as possible.

---

> > > ### Author Response · Authors · 2020-11-25
> > > **Response to Reviewer #3 (1/3)**
> > >
> > > **- "Thanks for clarifying and editing the paper - in the original version it seemed like both getting good intervals and estimating their coverage were equally important goals. It seems now that the main goal is estimating coverage. I think this is a somewhat less interesting goal since it's a more standard problem, but a thorough investigation of competing methods for this could be useful for the ICLR community."**
> > >
> > > First, we greatly thank the reviewer for the responses and the further comments.
> > >
> > > We want to emphasize that our goal is to get both good intervals and conditional coverage estimation simultaneously. These are indeed both equally important to us - If we don’t care about the high-quality criterion but only the conditional coverage, the problem, as the reviewer rightly suspects, would become less interesting (though still not entirely trivial). The main challenge here is the efficient incorporation of conditional coverage while ensuring the high-quality criterion, which motivates us to set up new assessment metrics and related loss functions in the training.
> > >
> > > To the best of our knowledge, this is the first paper that identifies and addresses such a challenge for PI, and its importance appears to be acknowledged by other reviewers as well. We suspect that the reviewer may think the conditional coverage estimation can be handled by known classification tools and so is a “more standard problem”, but this is not the case (as explained in detail in the response regarding “MSE” below), and one of our main contributions is to present this challenge and offer a remedying approach.
> > >
> > > **- 1 "If the main goal is to estimate coverage, then that's what the experiments should focus on. In the main paper, there is no comparison with alternative ways to estimate coverage. Table 2 only shows the ECE for the proposed method but no baselines. Appendix E.3 has a comparison on one dataset, but there are minimal details on the baseline approach, what hyperparameters were tuned, etc. Additionally, in the two stage method you don't need to train an entire new network from scratch to estimate coverage - you can just train a linear layer on top of the penultimate layer. Instead, the central experiment seems to be comparing the interval quality of other methods (Table 1), which from the author response seems like isn't the goal of the paper."**
> > >
> > > In Table 2 there is no baseline algorithm to compare with since, as explained above, we propose the first algorithm that incorporates conditional coverage estimation - and we hope our results can serve as a baseline for follow-up works in this important research line. The algorithm in Appendix E.3 is the decoupled version of our proposed model, and we use this comparison mainly to show the effectiveness of our end-to-end approach (as explained in our previous round of response, the end-to-end approach has advantages of conciseness and reducing the number of hyperparameters and neuron weights to be trained etc.). The reviewer’s suggestion for a “compact” two-stage algorithm is definitely reasonable - In fact, it is similar to our approach, but we are doing this in a more systematic way: Both the PI construction and our Ca-Module are connected to the penultimate layer so that they are output simultaneously, and associated with this is our loss function that accounts for both the PI marginal quality and conditional coverage. Compared with both the decoupled version and the reviewer’s suggested compact two-stage algorithm, our approach seems to be the most concise and it achieves good performances on both ends without the need of any extra training process.

---

### Public Comment · ~Gilad_Katz1 · 2020-11-11
**A few questions about the evaluation**

Dear authors,

I’ve read your paper with interest. As one who also studies this field, I find your approach interesting. While I was going over the evaluation results and their comparison to other leading studies (e.g., Quality-driven Prediction Intervals - QD), I had a few questions I was hoping you could find the time to answer:

1) Your evaluation is performed on datasets that are partially different from those used in all the other studies in the field (for example, Year Prediction MSD is missing). Do you have results for the missing datasets?

2) You seem to be using a different architecture from your baselines (i.e., 2 hidden layers) , but use the reported results of studies that only use one hidden layer. Would the baselines also benefit from the addition of another layer?

3) You report a 80/20 split, but your reported baseline results were done on a 90/10 split. Doesn't this cause a problem?

4) From the text I understand that one has to manually change the Lambda3 param for every dataset. This strikes me as very cumbersome. Isn’t there a single value that works well for all datasets like done in all other SOTA methods such as QD?

More generally, I like the approach and think the theoretical analysis is great. It would be very interesting to see how your approach scales to large datasets. The main reason I’m asking this is that your ECE1 evaluation metric is dependent on dataset size. Would this be problematic for large datasets?

Thanks in advance!

---

> ### Author Response · Authors · 2020-11-19
> **Thank you for your interest and recognition of our contributions!**
>
> Thank you for your interest and recognition of our contributions! We are glad that you like our approach and think our theoretical analysis is great. The following lists our brief answers to your questions, in which we also highlight the strengths of our method and study. You may find more details in the rebuttal to reviewers:
>
> **- 1. Our numerical experiments are based on the most widely used datasets.** PI is a popular topic with tons of available datasets (Kivaranovic et al., 2020, Zhu et al., 2019, Rosenfeld et al., 2018), and our selection already includes the most widely used datasets.
>
> **- 2. Our comparison is fair.** Pearce et al. uses different model architectures and hyper-parameters that we believe are optimized for their settings. Our optimal architecture and hyper-parameters are designed for our novel loss function. If we apply our architecture on Pearce et al., it may unfairly penalize their performances since the optimal architecture for our loss may not be optimal for them.
>
> **- 3. Our approach is competitive.** As discussed in Sections 2 and Appendix A.4, the test size should be sufficient in order to get a meaningful ECE evaluation, which motivates our split scheme. Moreover, even though our model is trained under less training data, it still gets comparable PI prediction results with baseline algorithms, which highlights the competitiveness of our approach.
>
> **- 4. Our algorithm is easy to implement.** Our tuning procedure for all hyper-parameters including $\lambda_3$, as documented in Appendix D, is transparent and easy to use.
>
> **- Our model achieves accurate and robust results on datasets with various dimensions and sizes.** We already evaluate our model on large datasets (the size of datasets ranges from 300 ~ 46000) and shows promising results. Moreover, we want to highlight that one important strength of our calibration-based error (equation (2.4)) is that its evaluation is free of the feature dimension, which means that ECE applies to large datasets and high dimensions as well.

---

### Author Response · Authors · 2020-11-19
**Paper Revision**

We sincerely thank all the reviewers for their thorough and constructive comments. We have detailed our individual response to each reviewer below. Here, we summarize the changes we have made in the paper as follows:

(1) [Clarifying goals] In Section 1, we have made explicit our goal to incorporate and assess conditional coverage estimation in high-quality PI construction (Reviewer #2, Reviewer #3).

(2) [Connections among different methodological aspects] In Section 3, we have added more explanations on the $L_{CA}$ loss to clarify its connection with (calibration-based) conditional coverage error in Section 2 and the theoretical analysis in Section 4 (Reviewer #2, Reviewer #3). We have separated the notations for the original version and the soft version of the CA-loss to improve readability (Reviewer #1). We have also highlighted more explicitly our theoretical contributions in Sections 1 and 4 (Reviewer #1).

(3) [Clarifying experimental details] We have added more explanations to clarify the criteria for empirically evaluating PIs in the caption of Table 1 (Reviewer #2). We have added the reasoning of our random split scheme in Appendix E.1 (Reviewer #1).

(4) [Comparisons with other methods] In Section 6 we have clarified that Bayesian approaches can also be used to construct PIs, and in Section 7 we have included their comparisons as our future direction (Reviewer #4).

(5) [Details of tuning procedures] We have added details of our tuning procedures for the hyper-parameters in Appendix D and Algorithm 1, to illustrate that it is transparent and easy-to-implement (Reviewer #1, Reviewer #3).

(6) [Advantages over the decoupled framework] We have included an additional section in the appendix (Appendix E.3) to show the advantages of our proposed end-to-end framework over a decoupled framework that first trains a PI predictor, then builds on and adjusts it to get conditional coverage estimation (Reviewer #1, Reviewer #2).

---

### Decision · Program_Chairs · 2021-01-07
**Final Decision**

**Decision:**

Reject

**Comment:**

The paper studies the problem of estimating high quality prediction intervals for deep regression models. The paper argues that one (relatively under-studied) avenue to improve these intervals is to accurately estimate conditional coverage -- traditional PIs only reason about marginal coverage. The paper argues that in the presence of heteroskedastic errors or model mis-specification, conditional coverage can be dramatically different than marginal coverage. Concrete examples for each of these cases would be useful to establish the claim -- a synthetic experiment later in the paper illustrates the gap using heteroskedastic errors. The paper introduces a "Confidence Assessment" module that estimates the probability that the model's confidence interval is correct. In spirit, this is akin to learning a calibrated probabilistic classifier. Theoretical analysis shows that the CA-module can provably assess the reliability of the confidence intervals while jointly training the confidence interval method -- some reviewers appreciated the rigor in this analysis.

However, the reviewers also pointed out that the main message of the paper is muddled, and the confusion spills over into the experimental execution of the paper. Many of the complaints about baselines and experiment setup can be traced back to this confusion.
There are several claims in the paper:
- Conditional coverage estimation is useful. The synthetic experiments demonstrate this sufficiently.
- The CA-module achieves conditional coverage estimation reliably and efficiently. There are missing baselines (e.g., other approaches implementing a probabilistic classifier) in the experiments to establish this claim. The authors added an experiment to address this, but reviewers are concerned that the baseline classifier is unnecessarily handicapped (e.g., training a new coverage model from scratch instead of using existing learned features). Reviewers also note that there are missing metrics -- the existing metrics can plausibly be gamed by simply outputting the marginal coverage estimate.
- Incorporating CA-module leads to better prediction intervals. Some experiments suggest that this is not the case, and that there is negligible improvement (the lambda_2 = 0 setting that the authors describe). On the other hand, it is heartening to note that adding the CA-module did not adversely affect the quality of the prediction intervals either.

Since a two-stage procedure (estimate intervals, followed by estimating CA-module) is empirically inferior to joint training, reviewers rightly ask for some insight into why estimating conditional coverage jointly would reliably lead to prediction intervals that are more precise on average. The theoretical analysis in the paper applies to the 2-stage procedure too (proving that the 2nd stage CA-module indeed estimates the reliability of the confidence intervals); so there is some missing insight on why joint training could be beneficial.

A clearer message, making weaker claims and experiments that clearly back those claims will make the paper stronger.
For example, (softening claims about CA-module:) The paper introduces one ad-hoc procedure (CA-module) and shows that it is fit for purpose. No claim that it is efficient relative to baselines, but it still needs to justify why CA-module should be preferred compared to any other probabilistic classification approach. (softening claims about better intervals:) joint training works better than stage-wise training (which, by definition, leaves the prediction intervals unaffected). Unclear as to why that should happen in general; two special cases are mis-specification and heteroskedasticity.